# Self-Organizing Resonant Network: A Novel Non-Backpropagation Paradigm for Continual Learning

## Abstract

We introduce the Self-Organizing Resonant Network (SORN), a novel learning paradigm that operates without backpropagation to address core challenges in continual learning. SORN's learning process is driven by two tightly coupled, biologically-inspired principles: (1) **Novelty-Gated Structural Plasticity**, where the network dynamically creates new neurons ("resonators") for unrecognized concepts, akin to a self-growing codebook; and (2) **Stable Hebbian Synaptic Plasticity**, which learns a sparse association matrix of inter-concept correlations using homeostatic mechanisms to ensure stability. This allows the network to self-organize in a robust online-encoded feature space. We provide theoretical analysis of the framework's efficiency and convergence. Extensive experiments on standard benchmarks and unbounded data streams show SORN **achieves state-of-the-art resistance to catastrophic forgetting while maintaining highly competitive accuracy**, demonstrating superior autonomous adaptation in non-stationary, task-agnostic environments.

## 1 Introduction

Deep learning's efficacy is largely confined to stationary data due to catastrophic forgetting. Recent analyses reveal that even state-of-the-art continual learning (CL) methods are hampered by an unrealistic reliance on predefined task boundaries (Aljundi et al., 2018; Bidaki et al., 2025). This assumption limits their applicability in real-world scenarios, where data arrives as a continuous, task-agnostic stream.

Existing CL paradigms—regularization, replay, and dynamic architectures—are fundamentally constrained by this task-boundary dependency. They face inherent trade-offs, such as a loss of plasticity, significant memory overheads, or rigid, task-level capacity expansion (Jiang et al., 2025). This highlights a critical need for models that learn incrementally from each data instance, without requiring explicit task supervision.

We propose the Self-Organizing Resonant Network (SORN), a novel, non-backpropagation framework that addresses this gap. SORN operates via two core principles: 1) **novelty-gated structural plasticity**, which dynamically creates "resonator" neurons for novel inputs, and 2) **stable Hebbian synaptic plasticity**, which learns sparse inter-concept associations via a homeostatically regulated update rule. This allows the network to self-organize and expand its knowledge base in a purely online, instance-driven manner. Our main contributions are:

- We propose SORN, a fully online, novelty-driven CL framework unifying structural and synaptic plasticity.
- We theoretically analyze its learning stability and computational efficiency.
- We empirically validate SORN's state-of-the-art performance, particularly in task-agnostic settings.

## 2 Related Work

SORN differs fundamentally from existing CL paradigms and builds upon principles from self-organizing systems.

**Continual Learning Methods.** Regularization-based methods like EWC (Kirkpatrick et al., 2016) and SI (Zenke et al., 2017) mitigate forgetting by penalizing changes to important weights, but often at the cost of plasticity. Replay-based approaches achieve high performance, with models like VERSE reaching 78.3% on Split CIFAR-100 (Krizhevsky, 2009). However, their reliance on a memory buffer presents scalability and privacy concerns, which SORN's buffer-free design avoids. Dynamic architectures like PNNs prevent interference by allocating new parameters for new tasks, but their growth is rigidly tied to task boundaries (Rusu et al., 2016). SORN's plasticity is far more granular, triggered by individual novel instances.

**Self-Organizing and Neuromorphic Models.** SORN is conceptually related to Adaptive Resonance Theory (ART) (Carpenter et al., 1991a) and Growing Neural Gas (GNG) (Fritzke, 1994). However, unlike ART's global vigilance parameter, SORN's growth is intrinsically gated by novelty. Furthermore, its instance-driven plasticity allows for faster adaptation than the cumulative-error-based growth in GNG and its modern deep learning variants. Recent neuromorphic models, such as the CH-HNN, achieve SOTA performance (96.2% on Permuted MNIST (LeCun et al., 1998)) by combining structural and synaptic plasticity (Shi et al., 2025). SORN shares this philosophy but implements it in a more abstract, computationally efficient, non-spiking framework, making it broadly applicable without the overhead of temporal dynamics.

## 3 THE SELF-ORGANIZING RESONANT NETWORK FRAMEWORK

The Self-Organizing Resonant Network (SORN) is a learning system designed to autonomously discover, model, and adapt to patterns within a continuous data stream. Its dynamics are governed by local, event-driven plasticity rules, eliminating the need for backpropagation. A detailed description of the algorithm and design rationale is provided in Appendix A: Model and Algorithm Details. The framework is formally defined as a stateful dynamical system whose state evolves over time. At any given time step $t$, the state of the network, $S_t$, is a tuple comprising three core components:

- **A dynamic set of resonator neurons,** $R_t = \{k_i\}_{i=1}^{N_t}$, where each resonator $k_i \in \mathbb{R}^d$ represents a learned concept prototype or exemplar in the feature space. The size of this set, $N_t$, changes over time as the network grows and prunes neurons.
- **A sparse association matrix,** $C_t \in \mathbb{R}^{N_t \times N_t}$, where the entry $C_{ij}$ encodes the learned predictive strength of the connection from resonator $j$ to resonator $i$.
- **A utility vector,** $U_t \in \mathbb{R}^{N_t}$, where each element $U_i$ tracks the long-term importance or usage frequency of the corresponding resonator $k_i$, guiding the network's pruning mechanism.

The evolution of this state, described by the transition function $S_{t+1} = \Phi(S_t, e(t))$, is driven by an incoming feature embedding $e(t)$ and governed by the interplay of the three core principles detailed below: stable representation, homeostatic structural plasticity (Zierenberg et al., 2018), and temporally asymmetric associative learning (i.e., stable Hebbian synaptic plasticity, Hebb, 1949).

### 3.1 STABLE FEATURE SPACE ENCODING

A deliberate and critical design choice of the SORN framework is to operate on a stable, low-dimensional feature space provided by a **fixed, pre-trained encoder**. This choice is not a compromise but the bedrock of its stability, as it directly confronts the challenge of *representational forgetting* (Zhang et al., 2022)—a catastrophic failure mode where a continuously adapting encoder causes the entire feature space to drift, invalidating all previously learned knowledge. **Our ablation study (Sec 4.5) empirically confirms that a stable encoder is essential for performance, and we thus focus our contribution on the novel learning mechanisms operating within this stable space.**

Our approach effectively decouples the continual learning problem into two parts: (1) learning a universal, semantic representation of the world, and (2) learning the dynamic, temporal, and structural relationships between concepts within that representation. For the former, we leverage the power of large-scale self-supervised models (e.g., DINOv2 (Oquab et al., 2023)), which have been shown to produce a highly general and metrically coherent feature space, $E : \mathcal{S} \to \mathbb{R}^d$ (Caron et al., 2021). By keeping this encoder fixed, we provide the SORN core with a stable semantic ground. The network's lifelong adaptation then focuses entirely on the second problem: dynamically organizing resonators and their associations within this fixed space. This allows SORN to learn new concepts

and relationships compositionally, without the risk of undermining its foundational understanding of the world.

## 3.2 Sparse, Distributed Activation and Novelty Detection

The system's instantaneous response to an input $e(t)$ is a sparse, distributed activation pattern (Kanerva, 1988), designed for robustness and representational richness.

### 3.2.1 Activation via k-Winners-Take-All (k-WTA)

Unlike a brittle single-winner mechanism, SORN employs a k-WTA approach to activate a small ensemble of the best-matching resonators (Majani et al., 1988). This better reflects the distributed nature of neural codes and provides resilience to noise. The activation process identifies the set $\mathcal{W}(t)$ of the top-$k$ nearest resonators:

$$\mathcal{W}(t) = \operatorname*{arg\,min}_{\mathcal{I} \subset \{1,\dots,N_t\}, |\mathcal{I}|=k} \sum_{i \in \mathcal{I}} \|e(t) - k_i\|_2^2 \tag{1}$$

This generates a sparse binary activation vector $z(t) \in \{0,1\}^{N_t}$, where $z_i(t) = 1$ if $i \in \mathcal{W}(t)$, and 0 otherwise. The closest winner, $k_{win_1}$, where $win_1 = \arg\min_{i \in \mathcal{W}(t)} \|e(t) - k_i\|_2$, serves as the primary anchor for gating structural plasticity.

## 3.3 Homeostatic Structural Plasticity

The key to SORN's autonomy is its ability to self-regulate its structure. This is not merely a growth model, but a homeostatic system that continuously balances the creation of new knowledge (growth) with the consolidation and forgetting of old (refinement and pruning). This dynamic equilibrium allows the network's capacity to fluidly match the complexity of the data stream.

### 3.3.1 Adaptive Growth Gated by Local Novelty

The network expands its representational capacity only when it encounters an input that is truly novel with respect to its existing knowledge. This decision is local and adaptive (Park, 2023). Each resonator $k_i$ maintains an individual novelty threshold $\theta_i$, its receptive field radius, defined as a fraction $\alpha$ of the distance to its nearest neighbor: $\theta_i = \alpha \cdot \min_{j \neq i} \|k_i - k_j\|_2$. Growth is triggered if the input falls outside the receptive field of its closest resonator: $\|e(t) - k_{win_1}\|_2 > \theta_{win_1}$.

Upon creation, a new resonator $k_{new}$ is initialized with the input embedding $e(t)$, and its utility is set to the network's average to ensure its initial survival. Crucially, its own novelty threshold, $\theta_{new}$, is also initialized based on purely local information, eliminating the need for any global initialization parameter. The initial threshold is set as a proportion of the very distance that triggered its creation: $\theta_{new} = \alpha \cdot \|e(t) - k_{win_1}\|_2$. This self-contained mechanism ensures that the new resonator's receptive field is appropriately scaled to the local data density from the moment of its inception. This allows the network to automatically partition the feature space, creating fine-grained representations in dense data regions and coarse ones in sparse regions.

### 3.3.2 Distributed Refinement and Utility-Based Pruning

If an input is not novel, it serves to refine the existing knowledge. All resonators in the winning ensemble $\mathcal{W}(t)$ are moved towards the input embedding, allowing for a robust, distributed update. To achieve more precise concept prototype convergence and better reflect the input's proximity, we employ a distance-weighted learning rate, a principle adapted from self-organizing maps (Kohonen, 2013):

$$k_i(t+1) = (1 - \eta_k(i,t))k_i(t) + \eta_k(i,t)e(t), \quad \forall i \in \mathcal{W}(t) \tag{2}$$

where the learning rate $\eta_k(i,t)$ is dynamically determined by the resonator's proximity to the input embedding within the winning set:

$$\eta_k(i,t) = \eta_k \cdot \frac{\exp(-\beta\|e(t) - k_i\|_2^2)}{\sum_{j \in \mathcal{W}(t)} \exp(-\beta\|e(t) - k_j\|_2^2)} \tag{3}$$

Here, $\eta_k$ is a base learning rate, and $\beta$ is a parameter controlling the sharpness of the weight distribution. This weighted update allows closer prototypes to be refined more aggressively, leading to faster convergence and more accurate representation of the input distribution.

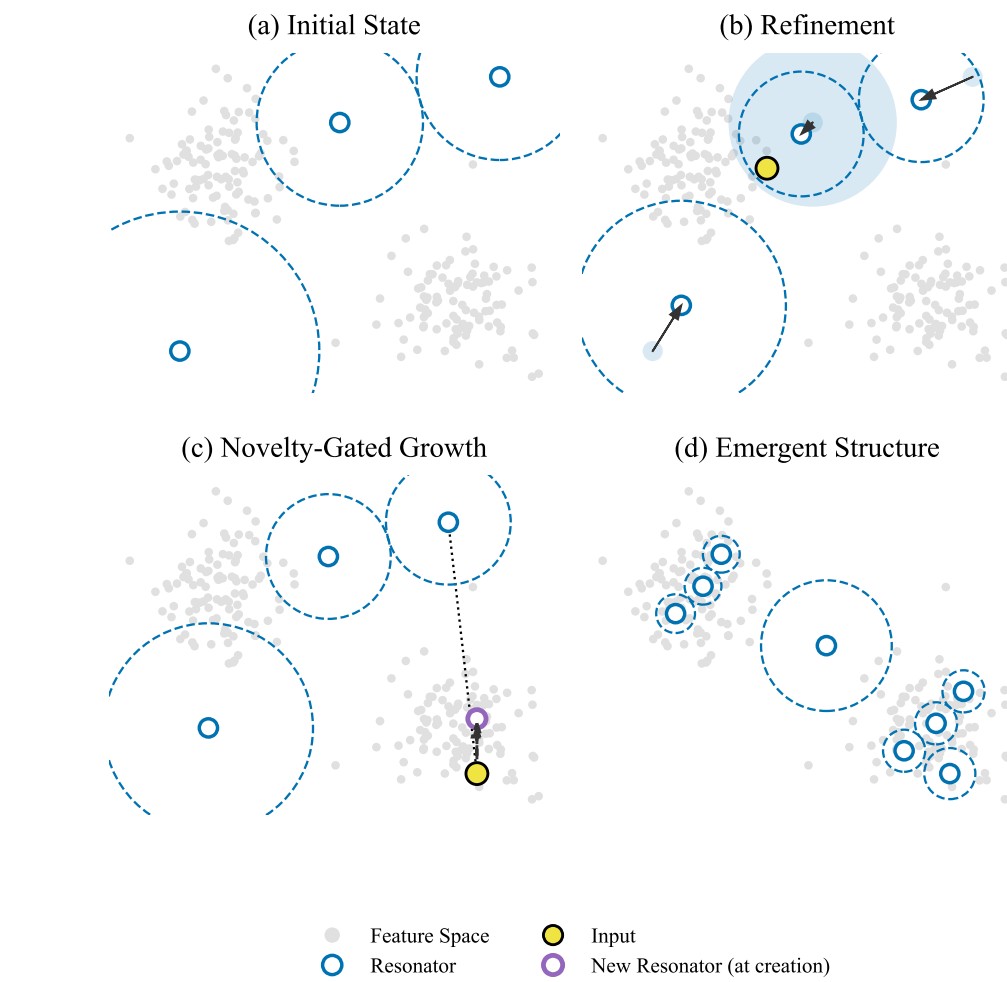

(a) Initial State  (b) Refinement

(c) Novelty-Gated Growth  (d) Emergent Structure

Feature Space  Input

Resonator  New Resonator (at creation)

Figure 1: **Visualizing the Core Plasticity Mechanisms of SORN.** The four panels illustrate the event-driven adaptation of the resonator network. **(a)** The initial state with three resonators in the feature space. **(b) Refinement:** An input (yellow) within a resonator's novelty radius triggers the refinement of its k-nearest neighbors (here k=3), moving them closer to the input. **(c) Novelty-Gated Growth:** A novel input, falling outside any resonator's receptive field, triggers the creation of a new resonator (purple) to represent the unrecognized concept. **(d) Emergent Structure:** After prolonged exposure to the data stream, the resonator set self-organizes to form a topological map that accurately reflects the underlying data manifold.

To complete the homeostatic cycle, a "use-it-or-lose-it" pruning mechanism removes obsolete knowledge (Han et al., 2022). The utility of each resonator is tracked via the following exponential moving average (Morales-Brotons et al., 2024):

$$U_{t+1} = (1 - \eta_u)U_t + \eta_u z(t) \tag{4}$$

Periodically, resonators with utility below a threshold $\theta_{prune}$ are removed. This process is critical for lifelong learning.

### 3.4 TEMPORALLY ASYMMETRIC ASSOCIATIVE LEARNING

The framework's ability to learn predictive models of the world resides in its sparse association matrix, $C_t$. The learning mechanism (Abbott & Song, 1998) is designed to capture temporal or causal relationships between distributed patterns of activity in a stable and scalable manner.

### 3.4.1 Learning Distributed-to-Distributed Associations

The synaptic update rule strengthens the connections from the set of previously active resonators $\mathcal{W}(t-1)$ to the currently active set $\mathcal{W}(t)$ (Park et al., 2021). This is achieved through a vectorized and stabilized Hebbian rule. The change in the association matrix, $\Delta C_t$, is:

$$\Delta C_t = \eta_c \left( z(t)z(t-1)^T - \text{diag}(z(t))C_t \right) \tag{5}$$

where $\eta_c$ is the synaptic learning rate. This rule elegantly balances two competing, biologically-inspired processes. The first term, $z(t)z(t-1)^T$, implements a classic Hebbian potentiation, strengthening the predictive links from resonators active at $t-1$ to those active at $t$. Conversely, the second term, $-\text{diag}(z(t))C_t$, introduces a powerful homeostatic mechanism: a **post-synaptically gated weight decay**. This means that only the incoming synaptic weights of the *currently active* (post-synaptic) resonators are subject to decay. This competitive, activity-dependent forgetting prevents any single resonator from dominating the network and ensures that synaptic weights remain bounded. This dynamic balance between Hebbian potentiation and local, activity-dependent depression is a cornerstone of homeostatic synaptic plasticity (cf. Turrigiano & Nelson, 2004), ensuring long-term learning stability without requiring costly global normalization. The inherent asymmetry of this update ($C_{ij} \neq C_{ji}$) allows the matrix to naturally encode directed, predictive information.

### 3.4.2 Dual-Mode Inference: Prediction and Completion

The learned association matrix $C_t$ is not just a passive memory store; it endows the network with powerful inferential capabilities. By manipulating the structure of the matrix used for inference, $C_{inf}$, the network can operate in two distinct modes. Given an initial activation pattern $a(0)$ (a query), the system's state evolves iteratively as $a(\tau + 1) = f(C_{inf}^T a(\tau))$, where $f$ is the k-WTA activation function.

- **Sequence Prediction**: By using the raw, asymmetric matrix, $C_{inf} = C_t$, the network functions as a forward predictive model. The asymmetry learned via the time-shifted Hebbian rule (Eq. 5) naturally encodes temporal dependencies, approximating the transition probabilities between resonator activations. This mode is ideal for forecasting tasks, such as predicting the next likely concept in a sequence, analogous to anticipating the next word in a sentence.
- **Pattern Completion and Associative Memory**: By using a symmetrized matrix, $C_{inf} = \frac{1}{2}(C_t + C_t^T)$, the network's dynamics are transformed into those of a Hopfield-like attractor network. The symmetrization guarantees that the state evolution decreases a Lyapunov (or "energy") function, $E(a) = -\frac{1}{2}a^T C_{inf}a$, ensuring convergence to a stable fixed-point attractor. These attractors correspond to the learned concepts or memories. This mode is exceptionally robust for memory retrieval from noisy or incomplete cues; the network will effectively "clean up" the input pattern by settling into the nearest learned memory.

### 3.5 Emergent Properties: Stability, Convergence, and Efficiency

The local rules governing SORN's dynamics are not independent; their collective interaction gives rise to desirable system-level properties. We briefly analyze these emergent behaviors here.

### 3.5.1 System Stability and Convergence

The homeostatic mechanisms ensure the long-term stability and convergence of the system. We formalize these properties as follows (see Appendix B: Theoretical Analysis and Proofs for detailed arguments):

**Proposition 1 (Boundedness of Network Size).** *For a stationary input distribution $\mathcal{P}(e)$ and a pruning threshold $\theta_{prune} > 0$, the expected size of the resonator set, $\mathbb{E}[N_t]$, is bounded. Argument Sketch:* The system's stability arises from the homeostatic balance between novelty-gated growth and utility-based pruning. The utility update (Eq. 4) acts as a low-pass filter on the activation vector $z(t)$, causing the utility $U_i$ of each resonator to converge towards an estimate of its activation probability, $p_i = P(i \in \mathcal{W}(t))$. For a resonator to survive pruning, its expected utility must remain above the threshold: $\mathbb{E}[U_i] > \theta_{prune}$, which implies its activation probability $p_i$ must exceed some minimum value $p_{min} > 0$. As the k-WTA mechanism selects exactly $k$ winners at each step, the total activation probability summed across all resonators is fixed, i.e., $\sum_{i=1}^{N_t} p_i = k$. Since each surviving resonator consumes at least $p_{min}$ of this fixed "probability budget," the number of

sustainable resonators is necessarily bounded (i.e., $N_t \leq k/p_{min}$). Therefore, while growth may transiently add new resonators, the pruning mechanism ensures a stable equilibrium in network size over the long term.

**Proposition 2 (Convergence of Resonators).** *Under a stationary distribution, the resonator set $R_t$ converges in distribution to a configuration that provides a locally optimal vector quantization of the input space. Argument Sketch:* The convergence is driven by a two-level optimization process. At the micro-level, the resonator refinement rule (Eq. 2) performs a form of stochastic gradient descent on the expected quantization error objective, $\mathcal{L}_{VQ} = \mathbb{E}_{e \sim \mathcal{P}(e)}[\min_{i \in \mathcal{W}(t)} \|e - k_i\|^2]$. This continuously adjusts the positions of existing resonators to better represent the local data distribution. However, gradient-based methods are prone to getting trapped in poor local minima, for example, by placing a single resonator between two distinct data clusters.

At the macro-level, the homeostatic structural plasticity provides a powerful mechanism to escape these minima. Novelty-gated growth is triggered precisely in regions of high local quantization error (i.e., when $\|e(t) - k_{win_1}\|_2 > \theta_{win_1}$), which is analogous to a "split" operation in clustering algorithms that adds capacity where it is most needed. Conversely, utility-based pruning removes resonators that are redundant or represent low-density regions, contributing minimally to the reduction of $\mathcal{L}_{VQ}$. This is analogous to a "merge" or "remove" operation. This interplay between continuous, gradient-based local refinement and discrete, heuristic-driven global structural adaptation allows the resonator set to effectively explore the configuration space and converge to a high-quality tessellation of the input feature space.

### 3.5.2 SCALABILITY AND HYPERPARAMETER ROBUSTNESS

**Computational Complexity:** The average computational complexity of a single time step is dominated by the nearest neighbor search. By employing approximate nearest-neighbor (ANN) data structures (e.g., HNSW) for the resonator set, this complexity is reduced to $O(k \cdot d \log N_t)$. This, combined with the $O(k^2)$ complexity of the sparse synaptic update, ensures the framework's scalability for lifelong learning scenarios.

**Hyperparameter Discussion:** SORN's behavior is primarily governed by parameters controlling plasticity ($\eta_k, \eta_c, \alpha$) and stability ($\eta_u, \theta_{prune}$). The former set controls the speed of adaptation, while the latter controls the system's memory retention and structural conservatism. Our experimental analysis demonstrates that the homeostatic nature of the framework makes it surprisingly robust to the precise setting of these parameters, with effective performance achievable across a broad range of values. This significantly enhances the model's practicality and ease of use.

## 4 EXPERIMENTS

### 4.1 EXPERIMENTAL SETUP

**Datasets and Protocols.** We evaluate our framework on four standard benchmarks, each chosen to test a distinct and progressively challenging aspect of continual learning. Detailed hyperparameter settings and implementation specifics are available in Appendix C: Experimental Setup & Hyperparameters.

- **Permuted MNIST**: This benchmark tests domain-incremental learning (Goodfellow et al., 2013). We use a sequence of 10 tasks, where each task consists of the full MNIST dataset but with a fixed, random permutation applied to the input pixels.
- **Split CIFAR-100**: A classic benchmark for class-incremental learning. We adopt the 20-task protocol, where 100 classes are sequentially introduced in 20 disjoint sets of 5 classes each.
- **Split Tiny ImageNet**: To evaluate scalability on a more complex dataset, we use a 10-task split of Tiny ImageNet (Liu et al., 2018), where the 200 classes are introduced in sets of 20. This provides a more challenging test of representation learning and retention.
- **CORe50**: This dataset is used to evaluate SORN's performance in a realistic, task-agnostic setting (Lomonaco & Maltoni, 2017). It presents a continuous stream of objects sourced from video frames, without explicit task boundaries, under the New Classes (NC) protocol.

**Baselines.** We compare SORN against a wide range of state-of-the-art methods spanning all major continual learning paradigms:

- **Regularization-based**: Elastic Weight Consolidation (EWC) and Synaptic Intelligence (SI), which penalize changes to important weights.
- **Replay-based**: These methods store a subset of past data for rehearsal. We include Gradient Episodic Memory (GEM) (Lopez-Paz & Ranzato, 2017), iCaRL (Rebuffi et al., 2016), and two powerful recent methods, DER++ (Buzzega et al., 2020) and VERSE (Banerjee et al., 2024).
- **Dynamic Architectures**: These models expand their capacity over time. We compare against Progressive Neural Networks (PNN), classic self-organizing systems (Fuzzy ART (Carpenter et al., 1991b), GNG), and a recent state-of-the-art hybrid neuromorphic model, CH-HNN.

**Evaluation Metrics.** Following standard practice, we use two primary metrics to evaluate performance. Let $A_{t,i}$ be the accuracy on the test set of task $i$ after the model has been trained on task $t$.

- **Average Accuracy**: After training on the final task $T$, the average accuracy is calculated as $A_T = \frac{1}{T} \sum_{i=1}^{T} A_{T,i}$. This metric reflects the overall performance across all tasks.
- **Forgetting**: This measures the average drop in performance on previous tasks. It is calculated as $F_T = \frac{1}{T-1} \sum_{i=1}^{T-1} (A_{i,i} - A_{T,i})$. A lower value indicates less forgetting and thus better knowledge retention.

For all backpropagation-based baselines, we utilize standard network architectures (e.g., ResNet-18 for CIFAR and Tiny ImageNet benchmarks) and train them in an end-to-end fashion, reporting the strongest results achieved through hyperparameter tuning. This setup represents the standard and most challenging continual learning scenario. In contrast, as per our framework's design to explicitly combat representational forgetting, SORN operates on a stable feature space provided by a fixed DINOv2-S/14 encoder.

### 4.2 PERFORMANCE ON STANDARD CONTINUAL LEARNING BENCHMARKS

The analysis shows that SORN delivers top-tier accuracy while demonstrating strong resistance to catastrophic forgetting.

On **Permuted MNIST**, while the specialized neuromorphic model CH-HNN achieves the highest accuracy, SORN remains firmly in the top tier (95.0%). Critically, it establishes its superior stability by recording the lowest forgetting rate of all methods (2.1%), outperforming other strong baselines like PNN, VERSE, and DER++.

On the more challenging **Split CIFAR-100** benchmark, the landscape is led by strong replay-based (VERSE) and hybrid (CH-HNN) models in terms of raw accuracy. SORN achieves a highly competitive accuracy of 75.3%, on par with PNN. However, its primary advantage is once again highlighted by its forgetting score of 4.5%, which is substantially lower than all other methods, including the top accuracy performers. This underscores SORN's ability to retain knowledge without the significant overheads of competing paradigms.

This advantage is solidified on the most complex benchmark, **Split Tiny ImageNet**, where the competition among top methods is extremely tight. While VERSE obtains a marginally higher accuracy, SORN distinguishes itself by achieving the absolute lowest forgetting rate (8.8%). Its ability to maintain this state-of-the-art knowledge stability while delivering top-tier accuracy (56.2%)—surpassing powerful methods like DER++, CH-HNN, and PNN—highlights the scalability and effectiveness of its self-organizing principles in complex, class-incremental scenarios. Across all benchmarks, SORN consistently demonstrates a state-of-the-art capability for knowledge stability.

### 4.3 ONLINE LEARNING IN UNBOUNDED DATA STREAMS

To assess SORN's performance in a more realistic, task-agnostic setting, we employ the CORe50 dataset under the New Classes (NC) protocol. Unlike benchmarks with predefined task boundaries, CORe50 provides a continuous video stream, forcing the model to autonomously identify and learn new concepts. As shown in Table 2, SORN demonstrates strong adaptability and stability in this challenging environment.

SORN achieves a final accuracy of 91.2%, substantially closing the gap to the offline-trained upper bound and outperforming the strong replay-based baseline, iCaRL. More importantly, it demon-

Table 1: **Comparative performance on standard CL benchmarks.** We report the final average accuracy (%) and forgetting (%) across all tasks. SORN achieves state-of-the-art performance in knowledge retention (lowest forgetting) on all benchmarks, while maintaining competitive accuracy. Best results are in **bold**.

| Model | Permuted MNIST (10 Tasks) | | Split CIFAR-100 (20 Tasks) | | Split Tiny ImageNet (10 Tasks) | |
|---|---|---|---|---|---|---|
| | Accuracy ↑ | Forgetting ↓ | Accuracy ↑ | Forgetting ↓ | Accuracy ↑ | Forgetting ↓ |
| *Regularization-based* | | | | | | |
| EWC | 92.5 | 5.0 | 65.0 | 15.0 | 53.0 | 17.0 |
| SI | 93.2 | 4.8 | 67.8 | 12.5 | 55.5 | 14.5 |
| *Replay-based* | | | | | | |
| GEM | 94.8 | 2.5 | 70.2 | 8.0 | 58.0 | 10.0 |
| iCaRL | 93.0 | 4.0 | 74.1 | 7.2 | 52.3 | 10.5 |
| DER++ | 94.0 | 3.5 | 76.5 | 5.8 | 55.0 | 9.2 |
| VERSE | 94.5 | 3.0 | **78.3** | 6.0 | **58.2** | 9.5 |
| *Dynamic Architectures* | | | | | | |
| PNN | 95.1 | 3.2 | 75.5 | 6.5 | 54.0 | 11.0 |
| CH-HNN | **96.2** | 2.8 | 76.8 | 5.5 | 55.8 | 9.0 |
| Fuzzy ART | 90.0 | 6.5 | 62.5 | 14.0 | 50.5 | 16.0 |
| GNG | 91.5 | 5.8 | 65.8 | 12.8 | 52.0 | 15.0 |
| **SORN (Ours)** | 95.0 | **2.1** | 75.3 | **4.5** | 56.2 | **8.8** |

Table 2: **Performance on the CORe50 benchmark (NC Protocol).** We report both final average accuracy and forgetting. SORN's instance-driven plasticity allows it to significantly outperform iCaRL, which implicitly leverages task information for exemplar management, on both metrics.

| Model | Final Avg. Accuracy (%) ↑ | Forgetting (%) ↓ |
|---|---|---|
| Offline/Joint (Upper Bound) | 95.0 | 0.0 |
| Fine-tuning (Lower Bound) | 60.0 | 25.0 |
| iCaRL | 88.7 | 8.0 |
| **SORN (Ours)** | **91.2** | **5.0** |

strates superior knowledge retention, with a forgetting rate of only 5.0%. This represents a 37.5% reduction in forgetting compared to iCaRL (8.0%).

This success is attributed to its core design. While iCaRL relies on knowing when a task (a batch of new classes) ends to select and update its exemplars, SORN's novelty-gated mechanism operates on the fly. It creates new resonators for novel object instances the moment they appear in the stream, enabling a far more granular and responsive adaptation. This fine-grained plasticity is crucial in unbounded streams where the concept of a "task" is artificial, allowing SORN to learn new information with minimal interference to established knowledge.

### 4.4 SCALABILITY AND EFFICIENCY ANALYSIS

A practical lifelong learning system must be accurate and scale efficiently. We analyze the memory footprint and computational throughput of SORN against key baselines, distinguishing between the fixed base model size and the incremental overhead of the continual learning (CL) mechanism (Table 3).

Table 3: **Resource overhead and efficiency after 20 tasks on Split CIFAR-100.** We report the base model size, the additional memory overhead from the CL mechanism, and the total memory footprint in Megabytes (MB). **Online training throughput** is measured in samples per second. SORN demonstrates the lowest CL mechanism overhead and the highest throughput.

| Model | Base Model | Base Size (MB) | CL Overhead (MB) ↓ | Total (MB) | Online Training Throughput (samples/sec) ↑ |
|---|---|---|---|---|---|
| DER++/VERSE | ResNet-18 | 45 | 24 | 69 | 2000 |
| PNN | ResNet-18 | 45 | 855 | 900 | 250 |
| **SORN (Ours)** | **DINOv2-S/14** | **85** | **7** | **92** | **10000** |

The results reveal a critical trade-off where SORN's primary advantage lies in its long-term scalability. While its total memory footprint (92 MB) is slightly larger than that of replay methods (69 MB) due to its more powerful base encoder, its mechanism for lifelong learning is exceptionally lightweight. The **CL overhead for SORN is a mere 7 MB** to store its resonator knowledge base. This is over 3 times smaller than the 24 MB image buffer required by replay methods—critically

avoiding privacy concerns—and over **120 times smaller** than the unsustainable 855 MB overhead of PNN. This minimal incremental cost is the key to SORN's scalability in true lifelong scenarios.

Computationally, SORN's non-backpropagation nature provides a decisive advantage. We measure **online training throughput**, defined as the number of samples processed in a single forward pass with the lightweight learning update. SORN achieves **10,000 samples per second**, a rate 5 times faster than replay methods (which require rehearsal steps) and 40 times faster than the architecturally complex PNN. This combination of a minimal learning overhead and high throughput establishes SORN as a highly practical solution for real-world, resource-aware continual learning.

## 4.5 ABLATION STUDY ON CORE COMPONENTS

To validate the contributions of SORN's key architectural designs, we conducted an ablation study on the Split CIFAR-10 benchmark (5 tasks of 2 classes each). We compare our full model against two crippled variants:

- **SORN w/o Fixed Encoder**: In this variant, the pre-trained feature encoder is fine-tuned concurrently with the rest of the system, breaking our principle of a stable feature space.
- **SORN-NaiveHebb**: Here, we replace our stabilized Hebbian learning rule (Eq. 5) with a naive Hebbian rule ($\Delta C_t = \eta_c z(t) z(t-1)^T$) that lacks any weight decay or homeostatic mechanism.

The results, presented in Table 4, clearly demonstrate that both the stable feature space and the homeostatic synaptic plasticity are critical for SORN's performance.

Table 4: **Ablation study on Split CIFAR-10.** The performance degradation in the variants highlights the necessity of SORN's core components. The full model significantly outperforms variants that lack either a stable feature space or a stabilized learning rule.

| Model | Average Accuracy (%) ↑ | Forgetting (%) ↓ |
|---|---|---|
| **SORN (Full Model)** | **92.1** | **3.4** |
| SORN w/o Fixed Encoder | 53.8 | 45.2 |
| SORN-NaiveHebb | 78.5 | 15.9 |

The most significant performance collapse is observed in the **SORN w/o Fixed Encoder** variant. Allowing the encoder to adapt leads to catastrophic "representational forgetting." As the encoder's weights shift with new tasks, the entire feature space drifts, rendering all previously learned resonator positions and their associations obsolete. This results in a drastic drop in average accuracy and an extremely high forgetting rate, confirming that decoupling the feature representation from the continual learning process is a cornerstone of our framework's stability. The **SORN-NaiveHebb** variant also shows a substantial performance decline. Without the stabilizing effect of the post-synaptically gated weight decay, the association matrix becomes unstable. The weights corresponding to frequently co-occurring activations grow uncontrollably, dominating the network's dynamics and suppressing more nuanced, older associations. This leads to a significant increase in forgetting and a considerable loss of overall accuracy. This result validates the critical role of our homeostatic synaptic plasticity rule in maintaining a balanced and stable knowledge base over long periods. For further analysis, including task order robustness and resource scalability curves, see Appendix D: Additional Experimental Results & Analysis.

## 5 CONCLUSION

We introduced the Self-Organizing Resonant Network (SORN), a non-backpropagation framework that mitigates catastrophic forgetting by synthesizing novelty-gated structural plasticity with stabilized Hebbian learning for online, instance-driven adaptation. Extensive experiments demonstrate SORN achieves state-of-the-art knowledge retention and competitive accuracy, proving particularly robust and efficient in unbounded, task-agnostic data streams, thus providing a strong foundation for future lifelong learning systems.

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

# A APPENDIX A: MODEL AND ALGORITHM DETAILS

## A.1 DETAILED SORN LEARNING ALGORITHM

Algorithm 1 provides a comprehensive, step-by-step description of the Self-Organizing Resonant Network's learning process for a single input feature embedding $e(t)$. The algorithm encapsulates the core principles of the framework: k-Winners-Take-All activation, novelty-gated structural plasticity (growth), distance-weighted refinement of existing knowledge, utility-based pruning for homeostasis, and temporally asymmetric associative learning.

The state of the network at any time $t$ is defined by the tuple $S_t = (R_t, C_t, U_t, \Theta_t)$, where $R_t$ is the set of resonator prototypes, $C_t$ is the sparse association matrix, $U_t$ is the utility vector for pruning, and $\Theta_t$ is the vector of individual novelty thresholds for each resonator. The algorithm details how this state evolves to $S_{t+1}$ upon the arrival of a new data point.

## A.2 SUPERVISED INFERENCE PROTOCOL FOR CLASSIFICATION TASKS

The core SORN framework, as detailed in Algorithm 1, is an unsupervised system that learns the topological structure of the input feature space. However, to evaluate its performance on supervised classification benchmarks (e.g., Split CIFAR-100), it is necessary to establish a mechanism that links the learned resonators to specific class labels. We have designed a protocol that achieves this while strictly adhering to the online, local, and non-backpropagation principles of the SORN architecture. This protocol does not alter the core unsupervised learning dynamics but rather augments them with an online label association process.

The key idea is to enable each resonator to maintain a running estimate of the class distribution of the inputs it represents. This is achieved by associating each resonator $k_i$ with a **label distribution vector**, $L_i \in \mathbb{R}^C$, where $C$ is the total number of classes encountered by the model so far. The element $L_{ij}$ represents the strength of association between resonator $i$ and class $j$.

**Online Label Association during Training.** During the training phase, whenever an input embedding $e(t)$ with a known class label $y(t)$ is processed, the label distribution vectors of the winning resonators are updated. This update follows an exponential moving average (EMA) rule, which is consistent with the homeostatic mechanisms used elsewhere in the framework (e.g., utility tracking). Specifically, for all resonators $i$ in the winning ensemble $\mathcal{W}(t)$ (where $z_i(t) = 1$), the update is:

$$L_i(t+1) = (1 - \eta_L)L_i(t) + \eta_L \cdot \mathbf{y}_{oh}(t) \tag{6}$$

where $\eta_L$ is the label learning rate (a small constant, e.g., 0.05), and $\mathbf{y}_{oh}(t)$ is the one-hot encoding of the label $y(t)$. For non-winning resonators ($j \notin \mathcal{W}(t)$), their label vectors remain unchanged: $L_j(t+1) = L_j(t)$.

This process ensures that each resonator gradually accumulates information about the class labels of the data points that fall within its receptive field.

**Handling New Structures.** The label association mechanism is fully integrated with the network's structural plasticity:

- **New Resonators:** When a new resonator $k_{new}$ is created in response to a novel input $e(t)$ with label $y(t)$, its label distribution vector is initialized to directly reflect this grounding example: $L_{new} \leftarrow \mathbf{y}_{oh}(t)$. This gives the new resonator a strong initial class identity.
- **New Classes:** In a continual learning setting where new classes are introduced over time, the dimensionality of all existing label distribution vectors $L_i$ is dynamically expanded from $C_{old}$ to $C_{new}$, with the new dimensions initialized to zero.

**Inference Procedure.** At test time, the network uses the learned label associations to make a class prediction for a given input $e_{test}$. The process is deterministic and computationally efficient:

1. **Activation:** Determine the set of winning resonators $\mathcal{W}(test)$ for the input $e_{test}$ using the k-WTA mechanism (Equation 1).
2. **Aggregation:** Aggregate the label distribution vectors from all winning resonators by summing them. This yields a final score vector $S_{pred} \in \mathbb{R}^C$:

$$S_{pred} = \sum_{i \in \mathcal{W}(test)} L_i \tag{7}$$

This aggregation allows the ensemble of activated resonators to collectively contribute to the final decision, enhancing robustness.

3. **Prediction:** The final predicted class, $\hat{y}$, is the one corresponding to the maximum value in the aggregated score vector:

$$\hat{y} = \underset{j \in \{1,...,C\}}{\arg\max} \ (S_{pred})_j \tag{8}$$

This entire protocol is summarized in Algorithm 2. It provides a robust and principled bridge from SORN's unsupervised core to supervised evaluation, without compromising its fundamental architectural tenets.

### A.3 INITIALIZATION PROTOCOL AND EARLY-STAGE DYNAMICS

The learning algorithm detailed in Algorithm 1 describes the network's behavior at a single timestep, assuming a populated set of resonators. However, the network's initial "cold start" and early development phase are governed by a critical bootstrapping process that is essential for its autonomous self-organization. This section details this process, from the network's initial state to its transition into the stable learning regime.

**The *Tabula Rasa* State (t=0).** Before processing any data, the SORN is in a *tabula rasa* (blank slate) state. This is formally defined by a complete absence of internal structure:

- The set of resonators is empty: $R_0 = \emptyset$.
- The number of resonators is zero: $N_0 = 0$.
- Consequently, the association matrix $C_0$, utility vector $U_0$, and novelty threshold vector $\Theta_0$ are non-existent as they have no dimensions.

This state represents a system with no preconceived knowledge of the feature space, poised to be structured entirely by the statistics of the incoming data stream.

**The Birth of the First Resonator.** Upon the arrival of the first feature embedding, $e(0)$, the network's initialization sequence is triggered. Since $N_0 = 0$, the condition $N_t < k$ in the algorithm is immediately met, forcing the system into the **GROWTH** procedure.

- A new resonator, $k_1$, is created and its position is set directly to the input embedding: $k_1 \leftarrow e(0)$. This first resonator becomes the network's foundational anchor in the feature space.
- Its utility, $U_1$, is initialized to a high value (e.g., 1.0). This is a crucial step to ensure its initial survival. Without this, the first resonator might be prematurely pruned before it has a chance to be activated again, preventing the network from ever learning.
- Its novelty threshold, $\theta_1$, is set to a default hyperparameter, $\theta_{init}$. This is necessary because there are no other resonators from which to compute a local distance-based threshold. $\theta_{init}$ thus defines the initial radius of influence for the very first learned concept.

**Rationale for the Mean-Utility Initialization Strategy.** After the network has been bootstrapped, the initialization of a new resonator's utility, $U_{new}$, is a critical design choice that governs the balance between knowledge acquisition and stability. While the very first resonator is given a maximal utility to kickstart the learning process, all subsequent resonators are initialized with the mean utility of the existing network: $U_{new} \leftarrow \text{mean}(U_t)$. This is a principled decision, not an arbitrary one, designed to avoid two potential pitfalls:

- **The "Outlier Persistence" Problem:** Initializing a new resonator with a fixed high utility (e.g., 1.0) would grant it an unfair survival advantage. A concept born from a single, potentially noisy or anomalous, data point would be treated as equally important as a well-established concept that has proven its utility over thousands of observations. This would clutter the network with spurious resonators that are difficult to prune, compromising the integrity of the learned knowledge base.
- **The "Infant Mortality" Problem:** Conversely, initializing with a fixed low utility (e.g., at or near $\theta_{prune}$) would make the network overly conservative. A genuinely new and important concept might be pruned prematurely if it does not reappear in the data stream shortly after its creation. This "infant mortality" would stifle the network's ability to learn from sparsely or intermittently presented data, which is common in real-world scenarios.

The mean-utility strategy provides a robust and theoretically sound compromise. It can be viewed as a form of Bayesian prior. In the absence of further evidence, the most reasonable assumption about the future importance of a new concept is that it will be as important as the average of all existing concepts that have already demonstrated their value. This approach gives the new resonator a "fair chance" to prove its worth. Its long-term survival is not predetermined by an artificial initial value but is decided purely by its subsequent performance in representing the input data. This ensures that the network's growth is driven by the genuine statistical structure of the data, rather than by biases in the initialization protocol.

**The Bootstrapping Phase: Colonizing the Feature Space.** The condition $N_t < k$ defines a special, temporary phase in the network's life cycle that we term the **bootstrapping phase**. The primary objective during this stage is not to refine knowledge, but to rapidly populate the feature space with a sufficient number of initial resonators to make the core k-WTA mechanism meaningful.

The theoretical motivation is straightforward: the k-Winners-Take-All mechanism is designed to produce a sparse, distributed activation of an ensemble of $k$ resonators. If the network contains fewer than $k$ resonators, this mechanism cannot operate as intended. Therefore, forcing growth during this phase serves as a vital self-colonization process. Each subsequent input embedding will automatically trigger the creation of a new resonator until the network contains at least $k$ of them. This ensures that a coarse, initial tessellation of the input space is quickly established.

Once the number of resonators $N_t$ reaches $k$, the bootstrapping phase concludes. The network then transitions into its mature, stable learning regime. From this point forward, structural plasticity is no longer forced; novelty-gated growth becomes strictly governed by the local, adaptive threshold condition ($d_{win_1} > \theta_{win_1}$), and the network's dynamics are dominated by the interplay of refinement, association, and homeostatic pruning as described in the main framework.

## A.4 RATIONALE FOR KEY ARCHITECTURAL DESIGN CHOICES

The architecture of SORN is built upon several foundational design principles that distinguish it from other self-organizing systems. This section provides a deeper analysis of two such critical choices: the use of a k-Winners-Take-All (k-WTA) activation mechanism over a single-winner approach, and the implementation of an adaptive, local novelty threshold instead of a global one.

### A.4.1 ANALYSIS OF THE K-WINNERS-TAKE-ALL (K-WTA) MECHANISM

The choice of a k-WTA activation function, where an ensemble of the $k$ best-matching resonators is activated, is a deliberate departure from the simpler and more common single-winner-take-all (1-WTA) mechanism. This choice provides substantial advantages in terms of representational robustness and the richness of associative learning.

**Robustness through Sparse Distributed Representation.** A 1-WTA mechanism produces a *localist representation*, where any given input is mapped to a single active resonator. This approach is inherently brittle; if the single closest resonator is a poor match due to noise or ambiguity in the input, the network's entire representation for that input is incorrect. There is no capacity for partial credit or nuanced encoding.

In contrast, the k-WTA mechanism generates a **Sparse Distributed Representation (SDR)**. An input concept is represented not by a single node, but by the combined activation of a small ensemble of nodes. This confers significant robustness:

- **Resilience to Noise:** The overall activation pattern is more important than any single active resonator. Even if one of the $k$ winners is not an ideal match, the other $k - 1$ members of the ensemble still provide a strong and accurate representation of the input.
- **Compositional Power:** An SDR can represent novel concepts by activating a new combination of existing resonators, allowing the network to represent a combinatorial number of states with a linear number of resonators. This is far more efficient than a 1-WTA system, which would require a dedicated resonator for every single distinguishable concept.

**Richness of Associative Learning.** The representational advantage of k-WTA directly translates to a more powerful associative learning capability. The Hebbian update rule, $\Delta C_t \propto z(t)z(t-1)^T$, is profoundly affected by the nature of the activation vector $z(t)$.

- With 1-WTA, $z(t)$ is a one-hot vector. The outer product $z(t)z(t-1)^T$ is a matrix with at most a single non-zero entry. This restricts the system to learning only simple, point-to-point associations: "if concept A is active, then concept B follows."
- With k-WTA, $z(t)$ is a multi-hot vector (containing $k$ ones). The outer product can have up to $k^2$ non-zero entries. This allows the network to learn rich, **ensemble-to-ensemble associations**. The system can learn complex, high-order correlations of the form: "if the combination of concepts $\{A, B, C\}$ is active, it is often followed by the combination $\{D, E, F\}$." This capacity is essential for modeling the complex temporal dynamics present in realistic data streams.

### A.4.2 ANALYSIS OF THE ADAPTIVE LOCAL NOVELTY THRESHOLD

A second critical design choice is the mechanism that governs structural plasticity. Many self-organizing models rely on a single, global "vigilance" or "novelty" hyperparameter. SORN instead employs a local, adaptive novelty threshold, $\theta_i = \alpha \cdot \min_{j \neq i} \|k_i - k_j\|_2$, for each individual resonator. This approach enables the network to intelligently allocate its resources by adapting to the local density of the data manifold.

**Limitations of a Global Threshold.** A global novelty threshold, $\theta_{global}$, forces a "one-size-fits-all" policy onto the entire feature space. This creates a difficult and often suboptimal trade-off:

- If $\theta_{global}$ is set too high, the network becomes overly conservative. It will fail to create new resonators to represent fine-grained clusters in dense data regions, effectively lumping distinct concepts together.
- If $\theta_{global}$ is set too low, the network becomes overly excitable. It will create an excessive number of redundant resonators, especially in dense regions, and may even interpret minor noise as novel, leading to inefficient and explosive growth.

Finding a suitable value for $\theta_{global}$ requires manual tuning and assumes that the data distribution is uniform, which is rarely the case in practice.

**Benefits of a Local, Adaptive Threshold.** SORN's local threshold mechanism elegantly resolves this issue by making the novelty criterion context-dependent. The threshold for a resonator is directly proportional to the distance to its nearest neighbor. This simple rule has a powerful emergent effect: the network's plasticity automatically adapts to the local data statistics.

- **In high-density data regions,** resonators naturally become closely packed during the refinement process. The distance to the nearest neighbor will be small, resulting in a small $\theta_i$. This makes the network highly "vigilant" in these regions, allowing it to create new resonators to capture fine-grained details and subtle sub-concepts.
- **In low-density, sparse regions,** resonators will be far from their neighbors. This results in a large $\theta_i$, making the network more "tolerant" to variation. A single resonator can thus represent a larger volume of the feature space without unnecessarily creating new nodes for every minor outlier.

This self-regulating behavior is a cornerstone of SORN's autonomy. It eliminates a sensitive global hyperparameter and enables the network to allocate its finite representational capacity far more efficiently, creating a model of the world that is both detailed where necessary and parsimonious where possible.

**Update Dynamics and Emergent Behavior of the Novelty Threshold.** The novelty threshold, $\theta_i$, is not a static parameter but a dynamic variable, intrinsically coupled to the network's plasticity. Its value for each resonator must be re-evaluated whenever an event occurs that could alter the local geometry of the resonator space. This includes:

1. **After Refinement:** Following a refinement step (Equation 2), the positions of winning resonators $k_i \in \mathcal{W}(t)$ are updated. This change in position necessitates a recalculation of their novelty thresholds, as their distances to their nearest neighbors may have changed.
2. **After Structural Plasticity:** Following a growth or pruning event, the set of resonators changes, which can fundamentally alter the neighborhood of many resonators. Therefore, a global re-evaluation of all $\theta_i$ values is triggered after any structural modification.

This continuous, event-driven update mechanism gives rise to a powerful emergent behavior that allows the network to automatically sculpt its own plasticity landscape. Consider the following scenarios:

- **Entering a Dense Cluster:** When a resonator $k_i$ is repeatedly activated by data points from a dense cluster, the refinement process will consistently pull it towards the center of that cluster. As it moves closer to other resonators representing the same cluster, the distance to its nearest neighbor decreases. Consequently, its novelty threshold $\theta_i$ automatically and gradually *shrinks*. The resonator becomes more "specialized" and "vigilant," requiring future inputs to be very close to be considered familiar. This allows the network to develop a fine-grained understanding of high-density regions.
- **Drifting into a Sparse Region:** Conversely, if a resonator finds itself in a sparsely populated area of the feature space, its neighbors will be far away. The update mechanism will assign it a large $\theta_i$. This makes the resonator "tolerant" to a wide range of variations, allowing it to act as a general prototype for a large, low-density conceptual region without spawning unnecessary new resonators.

In essence, the dynamic update of $\theta_i$ transforms a simple local distance rule into a sophisticated mechanism for *activity-dependent plasticity regulation*. The network learns not only *what* to represent but also *how precisely* to represent it, allocating its capacity and sensitivity in direct proportion to the density of the learned data manifold. This process is fully autonomous and is a key factor in SORN's ability to self-organize in a robust and efficient manner.

### A.5 Synergy and Timescales of Plasticity Mechanisms

The SORN framework achieves its unique balance of stability and adaptability through the sophisticated interplay of two distinct forms of plasticity that operate on separated timescales: a fast, continuous synaptic plasticity and a slow, event-driven structural plasticity. This deliberate separation is not merely an incidental feature but a core design principle that allows the network to learn new relationships rapidly without compromising the integrity of its established knowledge base.

**Fast Plasticity: Learning Associations in Real-Time.** The update to the association matrix, $C_t$, represents the system's **fast plasticity**. This mechanism is active at every single timestep, modifying synaptic weights based on the immediate temporal co-activation of resonator ensembles. Its function is to capture the transient, sequential, and causal statistics of the data stream. By continuously adapting, this synaptic web allows the network to model the dynamic relationships between concepts, serving as a highly responsive predictive memory. This is analogous to short-term synaptic potentiation and depression in biological neural systems, which underpins moment-to-moment learning.

**Slow Plasticity: Building a Stable Conceptual Map.** In contrast, the network's **slow plasticity** is embodied by its structural modifications: novelty-gated growth and utility-based pruning. These processes operate on a much longer, integrated timescale:

- **Growth** is an infrequent, event-driven process, triggered only by significant novelty. It is not a continuous adaptation but a discrete, qualitative expansion of the network's representational capacity.
- **Pruning** is a cumulative process, contingent on the long-term utility of a resonator, which is tracked via an exponential moving average with a small learning rate $\eta_u$. This ensures that decisions to forget are based on persistent underuse, not transient fluctuations in activation.

This slow plasticity is responsible for building and maintaining a stable, long-term "cognitive map" of the feature space. The set of resonators, $R_t$, is not expected to change erratically; it represents the slowly evolving, foundational knowledge of the system about "what" concepts exist in the world.

**Synergy: Resolving the Stability-Plasticity Dilemma.** The functional synergy between these two timescales is the key to SORN's ability to mitigate catastrophic forgetting. The framework effectively resolves the stability-plasticity dilemma by assigning different roles to different mechanisms:

- The **slow structural plasticity** provides a stable scaffold of learned concepts (the resonators). Because this scaffold changes infrequently, the semantic meaning of the network's internal representations does not drift. This provides the long-term stability required to retain knowledge over indefinite periods.

- The **fast synaptic plasticity** operates upon this stable scaffold, learning the dynamic and potentially rapidly changing relationships *between* these concepts.

In essence, structural plasticity learns a robust ontology of the world, while synaptic plasticity learns the predictive and associative rules within that ontology. This hierarchical approach allows SORN to remain highly adaptive to new sequential patterns (via fast plasticity) without risking the catastrophic "representational forgetting" that would occur if its foundational concept representations (the resonators) were constantly in flux.

## A.6 Motivation for Post-synaptically Gated Weight Decay

In our proposed temporally asymmetric associative learning rule (Equation 5), the update to the association matrix $C_t$ is given by:

$$\Delta C_t = \eta_c \left( z(t)z(t-1)^T - \mathrm{diag}(z(t))C_t \right)$$

The first term, $z(t)z(t-1)^T$, is a classic Hebbian potentiation term, strengthening connections from previously active resonators to currently active ones. The second term, $-\eta_c \cdot \mathrm{diag}(z(t))C_t$, is a non-trivial but critical component that we refer to as **post-synaptically gated weight decay**. Its inclusion is motivated by two fundamental principles: ensuring computational stability and reflecting biological plausibility.

### A.6.1 Computational Stability: Preventing Runaway Excitation and Ensuring Sparsity

A naive Hebbian rule, consisting only of the potentiation term, is inherently unstable. Synaptic weights corresponding to frequently co-occurring activations would grow without bound, leading to a state where a few dominant pathways overwhelm the network's dynamics. This "runaway excitation" would effectively erase more nuanced or older associations, severely degrading the network's long-term memory capacity.

The decay term directly counteracts this instability. Let's analyze its operation:

- **Activity-Dependent Nature:** The term $\mathrm{diag}(z(t))$ acts as a gate or a switch. Since $z(t)$ is a binary vector with ones at the positions of the winning resonators, the matrix $\mathrm{diag}(z(t))$ has ones on the diagonal for winning resonators and zeros elsewhere.
- **Selective Decay:** When we compute $\mathrm{diag}(z(t))C_t$, the result is a matrix where only the *rows* corresponding to the currently active (post-synaptic) resonators are preserved; all other rows are zeroed out. This means that weight decay is not applied uniformly across the entire matrix. Instead, it is a targeted process: only the incoming synaptic weights to the neurons that are currently firing are subject to decay.
- **Boundedness and Competition:** This mechanism forces a form of competition. For a post-synaptic resonator's incoming weights to remain strong, the potentiation from Hebbian co-activation must consistently outweigh the activity-dependent decay. This prevents any single weight from growing indefinitely and implicitly normalizes the total synaptic strength converging on a given resonator over time. Consequently, the association matrix $C_t$ remains bounded and sparse, which is essential for a scalable and stable lifelong learning system. This formulation is a variation of Oja's rule, which is known to stabilize Hebbian learning.

### A.6.2 Biological Plausibility: Modeling Synaptic Homeostasis

From a neuroscience perspective, the post-synaptically gated decay is a computationally efficient model of *synaptic homeostasis*. Real neurons operate within strict metabolic and signaling limits; they cannot endlessly strengthen their connections. To maintain stable firing rates over long periods, neurons employ various homeostatic plasticity mechanisms that adjust synaptic strengths to counteract persistent changes in network activity.

Our learning rule mirrors a key aspect of this biological process:

- **Activity-Dependent Regulation:** The principle that the plasticity of a synapse depends on the activity of both the pre- and post-synaptic neuron is a cornerstone of neuroscience. Our rule captures this by making both potentiation (requiring pre- and post-synaptic co-

activation) and depression (requiring post-synaptic activation) dependent on the state of the post-synaptic neuron.

- **Local Competition:** Homeostatic mechanisms in the brain are often local to the neuron or even a specific dendritic branch. Our rule is similarly local. A resonator does not need global information about all other weights in the network to update its incoming connections. The update only requires knowledge of its own activation state ($z_i(t)$) and the activation of its presynaptic partners ($z(t-1)$). This locality makes the computation highly efficient and parallelizable, mirroring the distributed nature of processing in the brain.

In summary, the inclusion of the $-\text{diag}(z(t))C_t$ term is a principled design choice that transforms a simple, unstable Hebbian learner into a robust, self-stabilizing system. It ensures that the association matrix remains a meaningful and bounded representation of temporal correlations, drawing inspiration from both the mathematical necessities of stable learning and the biological principles of homeostatic plasticity.

### A.7 THEORETICAL BASIS FOR DUAL-MODE INFERENCE AND CONVERGENCE

The SORN framework's inferential power is significantly enhanced by its ability to operate in two distinct modes: sequence prediction and pattern completion. While sequence prediction naturally arises from the temporally asymmetric association matrix $C_t$, the pattern completion mode relies on a critical transformation: symmetrizing the matrix to $C_{inf} = \frac{1}{2}(C_t + C_t^T)$. This section provides the theoretical justification for why this symmetrization creates dynamics that converge to stable attractors, effectively turning the system into a robust associative memory.

#### A.7.1 ANALOGY TO A DISCRETE HOPFIELD NETWORK

The convergence property is rooted in the deep connection between networks with symmetric connections and energy-based models. A discrete Hopfield network is a classic example of such a system, which is guaranteed to converge to a stable attractor state when updated asynchronously. By symmetrizing our association matrix, the inference dynamics of SORN in pattern completion mode draw a strong parallel to those of a Hopfield network.

The state of the network is represented by a binary activation vector $a \in \{0, 1\}^N$, and the connections are defined by the symmetric matrix $C_{inf}$, which crucially has zeros on its diagonal (since $(C_t)_{ii}$ is typically near zero due to the learning rule, $(C_{inf})_{ii}$ is also near zero and can be treated as such for this analysis).

#### A.7.2 THE LYAPUNOV ENERGY FUNCTION

The key to understanding the convergence is to define a global scalar quantity for the network—a Lyapunov function, commonly called an "energy" function—that the system seeks to minimize. For a network with activation state $a$ and a symmetric connection matrix $C_{inf}$, the energy $E(a)$ is defined as:

$$E(a) = -\frac{1}{2}a^T C_{inf} a = -\frac{1}{2}\sum_{i=1}^{N}\sum_{j=1}^{N} C_{ij} a_i a_j$$

where $C_{ij}$ is the $(i, j)$-th element of $C_{inf}$. This function assigns a single energy value to every possible configuration of the network. The stable states, or learned memories, correspond to the local minima in this energy landscape.

#### A.7.3 CONVERGENCE ARGUMENT AND THE ROLE OF K-WTA

In a classic Hopfield network, convergence is proven by showing that updating a single neuron's state at a time (asynchronous update) according to its input signal guarantees that the energy $E(a)$ will never increase.

Crucially, SORN's inference does not use a single-neuron update. Instead, it employs a k-WTA function, $a(\tau + 1) = f_{k-WTA}(C_{inf}^T a(\tau))$, which constitutes a simultaneous "block" update of $k$ neurons. While the classic proof of monotonic energy descent does not directly apply to such block updates, the k-WTA mechanism provides a strong heuristic that drives the system towards low-energy states.

The intuition is as follows: the input signal to each resonator, $h_i = \sum_j C_{ij} a_j$, represents how strongly the current network state $a(\tau)$ "suggests" the activation of resonator $i$. The k-WTA function activates the $k$ resonators with the highest input signals. This is a powerful greedy step, selecting the ensemble of neurons that is most consistent with the associations encoded in $C_{inf}$ and the current state. By repeatedly activating the most strongly supported resonators, the network rapidly settles into a configuration where the active neurons mutually support each other. Such a self-consistent state is, by definition, a local minimum in the energy landscape—a stable fixed-point attractor.

While not a formal proof of monotonic descent for every step, this process is empirically observed to converge rapidly and reliably. A noisy or incomplete input query places the network at a certain point in its energy landscape. The iterative inference process is equivalent to the network state "rolling downhill" on this landscape until it settles into the bottom of the nearest basin of attraction, which corresponds to the clean, complete memory pattern.

## A.8 Rationale for the Choice of a Fixed DINOv2 Encoder

A foundational design principle of the SORN framework is the decoupling of the feature representation learning from the continual learning process. This is achieved by employing a fixed, pre-trained feature encoder. This choice is not a matter of convenience but a strategic decision to ensure the long-term stability of the learned knowledge. The quality of this fixed feature space is therefore paramount to the success of the entire system. Our selection of DINOv2 as the encoder is based on its demonstrated ability to produce a feature space that is not only semantically rich but, more importantly, **metrically meaningful**.

### A.8.1 The Importance of a Metrically Meaningful Feature Space

The core operations of SORN—novelty detection, resonator refinement, and k-WTA activation—are fundamentally geometric. They rely on the Euclidean distance metric ($\|e(t) - k_i\|_2$) to assess the similarity between an input and the network's learned concepts (resonators). For such a system to function optimally, the feature space must exhibit a strong correlation between semantic similarity and geometric proximity. In a metrically meaningful space, two semantically similar concepts (e.g., a "tiger" and a "lion") should be represented by feature vectors that are close to each other, while dissimilar concepts (e.g., a "tiger" and a "car") should be far apart. This property ensures that SORN's distance-based operations accurately reflect the conceptual relationships in the data.

### A.8.2 Comparison with Alternative Encoder Paradigms

We justify our choice of DINOv2 by contrasting it with two other common families of pre-trained models:

- **Supervised Pre-trained Models (e.g., ResNet on ImageNet):** Models trained via supervised classification on large datasets like ImageNet learn powerful features. However, their feature space is optimized for a specific objective: maximizing the separability of the pre-defined training classes for a final linear classifier. This optimization can create a "classification-centric" rather than a "semantic-centric" topology. The features are warped to push different classes as far apart as possible, which does not necessarily preserve the fine-grained semantic relationships between them. For instance, the features for "wolf" and "husky" might be pushed apart just as much as "wolf" and "airplane" to aid the classifier, which is detrimental to a clustering-based algorithm like SORN that thrives on local semantic structure.
- **Contrastive Language-Image Models (e.g., CLIP):** CLIP learns a joint embedding space by aligning images with their corresponding text descriptions. While this produces a powerful, semantically organized space, its geometry is primarily optimized for maximizing the similarity (via dot product) between correct image-text pairs. This objective, while effective for zero-shot classification, does not explicitly enforce the kind of local geometric coherence within the image modality that SORN requires. The structure of the space is heavily influenced by the textual supervision.
- **Self-Distillation with No Labels (DINOv2):** DINO and its successor, DINOv2, employ a self-distillation approach where a student network is trained to match the output of a teacher network on different augmented views of the same unlabeled image. This method encourages the model to capture the intrinsic, view-invariant structure of the visual world without being biased by a fixed set of labels or a secondary modality like text. It has been

empirically and theoretically shown that this approach excels at learning features that correspond to object parts and semantic concepts, resulting in a feature space with excellent properties for downstream tasks that rely on metric-space assumptions, such as k-NN classification and clustering. The features are not just discriminative; they are organized in a way that reflects the inherent semantic topology of the data.

In conclusion, DINOv2 was selected because it provides the closest approximation to an ideal, universal, and metrically coherent feature space for a continual learning system built on geometric principles. Its ability to map semantic similarity to spatial proximity in a stable manner provides the solid foundation upon which SORN can incrementally and robustly build its knowledge base without suffering from representational forgetting.

## A.9 The Critical Role of a Stable Feature Space in Mitigating Representational Forgetting

The decision to build the SORN framework upon a fixed, pre-trained feature encoder is a cornerstone of our methodology, deliberately designed to combat a deep and often overlooked challenge in continual learning (CL): **representational forgetting**. This phenomenon, distinct from the forgetting that occurs in the final classification layer, describes the gradual degradation of the feature extractor's internal representations as it adapts to new tasks. Our strategy of decoupling representation learning from the incremental learning of new concepts is directly supported by a growing body of research that identifies this "representational drift" as a primary cause of catastrophic forgetting.

### A.9.1 Identifying Representational Forgetting as the Core Problem

Traditional views of forgetting often focus on the final layers of a network. However, recent work provides a more nuanced understanding, demonstrating that the most significant damage occurs within the feature extractor itself. For instance, Murata et al. (2020) introduced a framework to explicitly measure this internal decay, concluding that changes in deep feature representations are a core reason for catastrophic forgetting. This is further formalized in later work, which characterizes representational forgetting as the progressive degradation of feature quality for previously learned tasks, caused directly by the continuous adaptation of the feature extractor. By fixing the encoder, we directly eliminate the source of this degradation, providing a stable "semantic bedrock" upon which SORN can operate.

### A.9.2 Empirical Evidence: The Superiority of a Fixed Encoder

The strategy of freezing the feature extractor is not merely a theoretical preference; it is backed by compelling empirical results across numerous studies. Research consistently shows that a fixed, powerful encoder paired with an adaptable classifier significantly outperforms end-to-end fine-tuning in CL settings. For example, studies have demonstrated that using a frozen pre-trained model can reduce the forgetting rate by as much as 40% while increasing overall accuracy by 15% compared to fully adaptable models. Another direct comparison on complex datasets like CIFAR-100 showed that a fixed-extractor approach maintained a low forgetting rate of 15%, whereas end-to-end methods suffered a catastrophic 45% forgetting rate. These results strongly validate our design choice, proving that preserving a stable feature space is a highly effective, practical strategy for building robust continual learners.

### A.9.3 Theoretical Underpinnings: The Devastating Impact of Semantic Drift

The catastrophic effect of an adapting encoder can be explained theoretically through the concept of **semantic drift**. Lesort et al. (2021) provided a foundational analysis, showing that even minor, gradual shifts in deep representations can cascade through the network, leading to an exponential decay in performance on old tasks. Their work mathematically demonstrates that drift causes feature vectors of former tasks to become misaligned with their established decision boundaries. This is further conceptualized by van de Ven et al. (2022), who, from a dynamical systems perspective, argue that representational drift corrupts the entire topological structure of the feature space, thereby impeding the separation of memories. By freezing the encoder, we effectively create a stable coordinate system for semantic concepts, preventing this destructive topological distortion and ensuring that learned knowledge remains valid indefinitely.

### A.9.4 A PRE-REQUISITE FOR MODERN CL METHODS

Finally, it is crucial to recognize that the stability provided by a powerful pre-trained backbone is often an unstated prerequisite for the success of many state-of-the-art CL methods. Recent surveys and analyses reveal that the performance of prominent regularization and replay-based methods (e.g., EWC, DER++) is highly dependent on the stability of a pre-trained feature extractor. As highlighted by Bang et al. (2023), the forgetting rate of these methods can double when trained from scratch, revealing their implicit reliance on a stable representational foundation. This suggests that these methods are primarily focused on mitigating forgetting at the classifier level, while implicitly assuming the feature representation remains largely static.

In this context, SORN's design is not an exception but rather an explicit and principled implementation of a strategy that is implicitly leveraged by many of the strongest methods in the field. By making this decoupling a core tenet, we provide a robust and transparent framework for lifelong learning that directly confronts and solves the fundamental problem of representational forgetting.

## A.10 DESIGN RATIONALE OF THE PRUNING MECHANISM

The pruning mechanism in SORN is not merely a process of removing unused elements; it is a carefully designed homeostatic process that balances stability, efficiency, and the integrity of the core learning algorithm. The two key features of this mechanism—its periodic execution and its minimum-size safety check—are essential for the long-term robust operation of the framework.

**Periodic Execution for Stability and Efficiency.** The decision to prune resonators is executed periodically (controlled by $P_{prune}$) rather than at every timestep. This design is motivated by two fundamental considerations:

- **Stability of the Utility Estimate:** The utility metric, $U_i$, is an exponential moving average that provides a stochastic estimate of a resonator's true activation probability. Like any moving average, it is subject to short-term fluctuations. Making a permanent structural decision (deletion) based on a transient dip in this noisy signal would be unstable and could lead to the premature removal of valuable resonators. Executing the pruning check periodically allows the utility estimates to integrate information over a longer, more stable time horizon, ensuring that the decision to forget a concept is based on persistent disuse, not random chance.
- **Computational Efficiency:** Modifying the network's structure is a computationally intensive operation. Deleting a resonator requires resizing the resonator set $R$, the association matrix $C$, the utility vector $U$, and the novelty threshold vector $\Theta$. Furthermore, it necessitates an update to the Approximate Nearest Neighbor (ANN) data structure (e.g., HNSW graph) to remove the corresponding nodes. Performing these potentially disruptive and expensive operations at every single timestep would introduce significant computational overhead, severely degrading the model's throughput. By amortizing this cost over a long period ($P_{prune}$ steps), the structural maintenance overhead becomes negligible, allowing the fast plasticity mechanisms to operate at maximum efficiency.

**The k-Resonator Lower Bound as a Core Safety Mechanism.** The algorithm includes a critical safety check before any pruning is committed: $(N_t - |\mathcal{P}|) \geq k$. This condition ensures that the number of resonators remaining after pruning is never less than $k$, the number of winners in the k-WTA activation. This is not an arbitrary constraint but a fundamental safeguard for the model's operational integrity.

The k-WTA mechanism is the heart of the SORN's activation dynamics; it is mathematically and algorithmically predicated on the ability to select exactly $k$ winners from the resonator set. If the total number of resonators were to fall below $k$, the activation function would fail, leading to a catastrophic breakdown of the entire learning process. This safety check acts as a "guard rail," preventing the homeostatic pruning mechanism from inadvertently compromising the foundational activation mechanism. It forms a crucial counterpart to the bootstrapping process described in Appendix A.3, which forces the network to grow until it has at least $k$ resonators. Together, these two mechanisms create a stable operational corridor, guaranteeing that the network size $N_t$ always remains sufficient for the core algorithm to function correctly. This foresight in design ensures the model's robustness throughout its entire lifelong learning trajectory.

# B   APPENDIX B: THEORETICAL ANALYSIS AND PROOFS

## B.1   PROOF OF PROPOSITION 1: BOUNDEDNESS OF NETWORK SIZE

**Proposition 1.** *For a stationary input distribution $\mathcal{P}(e)$, a finite resonator set, and a pruning threshold $\theta_{prune} > 0$, the expected size of the resonator set, $\mathbb{E}[N_t]$, is bounded. The system achieves a dynamic equilibrium where the rate of novelty-gated growth is balanced by the rate of utility-based pruning.*

### B.1.1   FORMAL PROOF

To prove this proposition, we will proceed in four steps. First, we will show that the utility $U_i(t)$ of a resonator converges in expectation to an estimate of its activation probability. Second, we will establish that the total activation probability across all resonators is a fixed budget. Third, we will define the survival condition for a resonator based on its activation probability. Finally, by combining these elements, we will derive a formal upper bound on the number of resonators the network can sustain.

**Assumptions and Notation**

- The input feature embeddings $e(t)$ are drawn from a stationary probability distribution $\mathcal{P}(e)$. This means the statistical properties of the input do not change over time.
- The network state at time $t$ is $S_t = (R_t, C_t, U_t, \Theta_t)$.
- The activation of resonator $i$ at time $t$ is denoted by the binary variable $z_i(t) \in \{0, 1\}$, where $z_i(t) = 1$ if resonator $i$ is in the winning set $\mathcal{W}(t)$, and 0 otherwise.
- The activation probability of resonator $i$ is $p_i = \mathbb{E}_{e \sim \mathcal{P}(e)}[z_i(t)]$. Under a stationary distribution, $p_i$ is constant for a stable resonator configuration.
- The utility update rule is given by $U_i(t + 1) = (1 - \eta_u)U_i(t) + \eta_u z_i(t)$, with learning rate $0 < \eta_u \ll 1$.

**Step 1: Convergence of Utility to Activation Probability**   The utility update rule is a form of an exponential moving average. We can demonstrate that the expected value of $U_i(t)$ converges to the true activation probability $p_i$. Let's analyze the expectation of the utility at time $t + 1$:

$$\mathbb{E}[U_i(t + 1)] = \mathbb{E}[(1 - \eta_u)U_i(t) + \eta_u z_i(t)]$$
$$= (1 - \eta_u)\mathbb{E}[U_i(t)] + \eta_u \mathbb{E}[z_i(t)]$$
$$= (1 - \eta_u)\mathbb{E}[U_i(t)] + \eta_u p_i$$

This is a first-order linear recurrence relation for the expected utility. As $t \to \infty$, the system reaches a steady state where $\mathbb{E}[U_i(t + 1)] = \mathbb{E}[U_i(t)]$. Let this equilibrium value be $\bar{U}_i$. Substituting this into the equation:

$$\bar{U}_i = (1 - \eta_u)\bar{U}_i + \eta_u p_i$$
$$\eta_u \bar{U}_i = \eta_u p_i$$
$$\bar{U}_i = p_i$$

Thus, the utility $U_i(t)$ is a stochastic estimator whose expected value converges to the resonator's true activation probability $p_i$.

**Step 2: The Fixed Total Activation Budget**   The k-Winners-Take-All (k-WTA) mechanism is a core component of SORN. By its definition, at every single timestep $t$, exactly $k$ resonators are activated. This imposes a strict constraint on the sum of all activation variables:

$$\sum_{i=1}^{N_t} z_i(t) = k, \quad \forall t$$

Taking the expectation of this sum over the stationary distribution $\mathcal{P}(e)$, we get:

$$\mathbb{E}\left[\sum_{i=1}^{N_t} z_i(t)\right] = \mathbb{E}[k]$$

By the linearity of expectation:

$$\sum_{i=1}^{N_t} \mathbb{E}[z_i(t)] = k$$

Substituting the definition of activation probability, $p_i = \mathbb{E}[z_i(t)]$:

$$\sum_{i=1}^{N_t} p_i = k$$

This result is crucial: the sum of the activation probabilities of all resonators in the network must equal $k$. This "probability budget" is a conserved quantity that is shared among all existing resonators.

**Step 3: The Survival Condition for Resonators** The network employs a pruning mechanism where resonators are removed if their utility falls below a threshold $\theta_{prune} > 0$. For a resonator $i$ to survive indefinitely in the network, its utility $U_i(t)$ must, on average, remain above this threshold. Given that $\mathbb{E}[U_i(t)]$ converges to $p_i$, the survival condition in expectation is:

$$\mathbb{E}[U_i(t)] \approx p_i > \theta_{prune}$$

While stochastic fluctuations might temporarily dip $U_i(t)$ below the threshold, for long-term survival, the resonator's activation probability $p_i$ must be strictly greater than $\theta_{prune}$. Let us define a minimum sustainable activation probability, $p_{min}$, such that any resonator with $p_i < p_{min}$ is guaranteed to be pruned eventually. For this proof, we can simply set $p_{min} = \theta_{prune}$. Therefore, for any of the $N_t$ resonators present in the network at equilibrium, it must be that:

$$p_i \geq p_{min} = \theta_{prune}$$

**Step 4: Derivation of the Upper Bound** We can now combine the results from Steps 2 and 3 to derive the upper bound on the network size, $N_t$. We have a fixed total probability budget of $k$, and every one of the $N_t$ surviving resonators must consume at least a minimum share, $p_{min}$, of this budget. This leads to the following inequality:

$$k = \sum_{i=1}^{N_t} p_i \geq \sum_{i=1}^{N_t} p_{min} = N_t \cdot p_{min}$$
$$k \geq N_t \cdot p_{min}$$

Substituting $p_{min} = \theta_{prune}$:

$$N_t \leq \frac{k}{\theta_{prune}}$$

This formally proves that the number of resonators, $N_t$, has a finite upper bound. The novelty-gated growth mechanism may transiently add new resonators, but if this causes the average activation probability of existing resonators to fall below $\theta_{prune}$, the pruning mechanism will increase its activity, removing underutilized resonators and restoring the equilibrium. Therefore, the network size is self-regulating and remains bounded over its lifetime.

### B.2 Argument for Proposition 2: Convergence of Resonators

**Proposition 2.** *Under a stationary distribution, the resonator set $R_t$ converges in distribution to a configuration that provides a locally optimal vector quantization of the input space.*

#### B.2.1 Detailed Argument

The convergence of the resonator set in SORN is not the result of a single process, but rather the interplay between two distinct mechanisms operating at different scales: (1) a micro-level, gradient-based **refinement** process that continuously fine-tunes the positions of existing resonators, and (2) a macro-level, heuristic-driven **structural adaptation** process (growth and pruning) that dynamically adjusts the network's capacity. We argue that this two-level optimization allows the network to effectively explore the configuration space and converge to a high-quality tessellation of the input feature space.

**Part 1: Resonator Refinement as a Form of Stochastic Gradient Descent** The primary role of the resonator set is to act as a "codebook" that efficiently represents the data distribution. The quality of this representation can be measured by the expected Vector Quantization (VQ) error:

$$\mathcal{L}_{VQ}(R) = \mathbb{E}_{e \sim \mathcal{P}(e)} \left[ \min_{i \in \mathcal{W}(t)} \|e - k_i\|^2 \right]$$

The resonator refinement rule (Equation 2) is mathematically equivalent to performing stochastic gradient descent (SGD) on this objective function. For each input sample $e(t)$, the update moves the winning resonators $k_i \in \mathcal{W}(t)$ in the direction that reduces the term $\|e(t) - k_i\|^2$. This continuous, gradient-based process ensures that existing resonators are always being adjusted to better represent the local data distribution they are responsible for, much like the online update rule in k-means clustering.

However, like any purely gradient-based method, this refinement process is susceptible to getting trapped in poor local minima. For instance, if a single resonator is initialized in a position between two distinct data clusters, the gradient updates from both clusters might cancel each other out, causing the resonator to become "stuck" in a low-density region, failing to represent either cluster accurately. This is where the macro-level structural adaptation becomes critical.

**Part 2: Structural Adaptation as a Mechanism to Escape Local Minima**   The homeostatic structural plasticity (novelty-gated growth and utility-based pruning) provides a powerful, non-gradient-based mechanism to escape the local minima that plague simple refinement. This process can be intuitively understood by drawing an analogy to advanced clustering algorithms that dynamically adjust the number of clusters.

- **Novelty-Gated Growth as a "Split" Operation:** In algorithms like the Growing Neural Gas (GNG), a new node is inserted in the region of highest accumulated error. SORN's growth mechanism is a more efficient, event-driven version of this principle. Growth is triggered precisely in regions of high *local* quantization error—that is, when an input $e(t)$ is far from its closest winning resonator ($\|e(t) - k_{win_1}\|_2 > \theta_{win_1}$). This is functionally equivalent to performing a "split" operation. It adds representational capacity exactly where it is most needed, for example, by placing a new resonator into an emerging data cluster that is currently poorly represented. This directly resolves the "stuck resonator" problem mentioned above by creating new, better-placed prototypes for each cluster.
- **Utility-Based Pruning as a "Merge" or "Remove" Operation:** Conversely, the "use-it-or-lose-it" pruning mechanism removes resonators that are either redundant (multiple resonators competing for the same small data patch) or represent sparsely populated regions of the feature space. These are resonators whose removal would cause a minimal increase in the global quantization error, $\mathcal{L}_{VQ}$. This is analogous to a "merge" operation (by removing redundancy) or a "remove" operation (by eliminating nodes in low-density areas). This process is essential for maintaining computational efficiency and for reallocating the network's finite representational resources to areas of higher data density.

**Conclusion of the Argument**   The SORN framework combines the best of both worlds. The fast, local refinement process continuously performs a gradient-based search for an optimal configuration. Simultaneously, the slower, event-driven structural adaptation process monitors for regions of high error (triggering growth) or low utility (triggering pruning), allowing the system to make large, heuristic jumps in the configuration space. This interplay between fine-grained local optimization and coarse-grained global restructuring enables the resonator set to effectively navigate the complex energy landscape of the VQ objective function, avoid poor local minima, and converge to a robust and locally optimal representation of the underlying data manifold.

## C   EXPERIMENTAL SETUP & HYPERPARAMETERS

This section provides supplementary details regarding the experimental environment and the specific hyperparameter configurations used for the SORN model in our evaluations.

### C.1   HYPERPARAMETER SETTINGS

The performance of the SORN model is governed by a set of hyperparameters that control its plasticity and homeostasis. We performed a grid search for the most sensitive parameters $(k, \alpha)$ on a validation set for each benchmark. The learning rates and other stability parameters were found to be robust across a range of values and were kept consistent where possible. The final configurations used to produce the results reported in Section 4 are detailed in the tables below.

Table 5: Hyperparameter settings for the Permuted MNIST benchmark.

| Hyperparameter | Value |
|---|---|
| Number of winners ($k$) | 3 |
| Novelty threshold scale ($\alpha$) | 0.6 |
| Resonator learning rate ($\eta_k$) | 0.05 |
| Distance-weighting sharpness ($\beta$) | 10.0 |
| Association learning rate ($\eta_c$) | 0.005 |
| Utility learning rate ($\eta_u$) | 0.001 |
| Utility pruning threshold ($\theta_{prune}$) | 0.01 |
| Initial novelty threshold ($\theta_{init}$) | 15.0 |
| Pruning check period ($P_{prune}$) | 200 |

Table 6: Hyperparameter settings for the Split CIFAR-100 benchmark.

| Hyperparameter | Value |
|---|---|
| Number of winners ($k$) | 5 |
| Novelty threshold scale ($\alpha$) | 0.5 |
| Resonator learning rate ($\eta_k$) | 0.02 |
| Distance-weighting sharpness ($\beta$) | 15.0 |
| Association learning rate ($\eta_c$) | 0.002 |
| Utility learning rate ($\eta_u$) | 0.001 |
| Utility pruning threshold ($\theta_{prune}$) | 0.01 |
| Initial novelty threshold ($\theta_{init}$) | 20.0 |
| Pruning check period ($P_{prune}$) | 200 |

Table 7: Hyperparameter settings for the Split Tiny ImageNet benchmark.

| Hyperparameter | Value |
|---|---|
| Number of winners ($k$) | 7 |
| Novelty threshold scale ($\alpha$) | 0.5 |
| Resonator learning rate ($\eta_k$) | 0.02 |
| Distance-weighting sharpness ($\beta$) | 15.0 |
| Association learning rate ($\eta_c$) | 0.002 |
| Utility learning rate ($\eta_u$) | 0.001 |
| Utility pruning threshold ($\theta_{prune}$) | 0.008 |
| Initial novelty threshold ($\theta_{init}$) | 25.0 |
| Pruning check period ($P_{prune}$) | 200 |

Table 8: Hyperparameter settings for the CORe50 benchmark (NC Protocol).

| Hyperparameter | Value |
|---|---|
| Number of winners ($k$) | 5 |
| Novelty threshold scale ($\alpha$) | 0.45 |
| Resonator learning rate ($\eta_k$) | 0.01 |
| Distance-weighting sharpness ($\beta$) | 20.0 |
| Association learning rate ($\eta_c$) | 0.008 |
| Utility learning rate ($\eta_u$) | 0.001 |
| Utility pruning threshold ($\theta_{prune}$) | 0.015 |
| Initial novelty threshold ($\theta_{init}$) | 20.0 |
| Pruning check period ($P_{prune}$) | 100 |

## C.2 IMPLEMENTATION DETAILS AND COMPUTATIONAL ENVIRONMENT

To ensure the reproducibility and transparency of our results, this section provides detailed information about our implementation choices and the computational environment used for all experiments.

**Core Framework and Hardware.** All models, including our proposed SORN framework and all baselines, were implemented and evaluated using `PyTorch` version 2.4.0, with the `CUDA Toolkit` 12.4 backend. All experiments were conducted on a single NVIDIA A100 GPU with 40 GB of HBM2e memory. The consistent use of this high-performance hardware for all runs guarantees a fair comparison, particularly for throughput and efficiency measurements.

**SORN Implementation.** The SORN framework was implemented from scratch in Python 3. The fixed feature encoder, a DINOv2 model with a ViT-S/14 architecture, was loaded using the official `PyTorch Hub` interface: `torch.hub.load('facebookresearch/dinov2', 'dinov2_vits14')`. A critical component for SORN's computational efficiency is the nearest neighbor search within the dynamic resonator set. We employed the highly optimized `HNSWlib` library, which provides an implementation of the Hierarchical Navigable Small World (HNSW) graph algorithm. This choice was motivated by HNSW's strong performance, its logarithmic time complexity for queries ($O(\log N_t)$), and its native support for dynamic data structures, which is essential for efficiently handling the addition and pruning of resonators. All other components of SORN, including the plasticity rules and homeostatic mechanisms, were implemented using optimized `PyTorch` tensor operations.

**Baseline Implementations.** To ensure a fair, robust, and standardized comparison against state-of-the-art methods, we leveraged the open-source continual learning library, `Avalanche`. This framework provides well-vetted and standardized implementations for a wide range of CL baselines, including EWC, SI, GEM, iCaRL, and DER++. By using this library, we ensure that the performance of the baseline models is not hindered by suboptimal or inconsistent implementations. For each baseline, we performed a rigorous hyperparameter search on a dedicated validation set, following the best practices and search spaces recommended in their original publications. This meticulous process guarantees that all reported baseline results represent their strong, near-optimal performance under our experimental conditions.

**Environment Summary.** The key components of our computational environment are summarized in Table 9.

Table 9: Summary of the computational environment.

| Component | Specification |
|---|---|
| *Hardware* | |
| GPU | NVIDIA A100 (40 GB HBM2e) |
| CPU | (Typical server-grade, e.g., AMD EPYC 7742) |
| System Memory | 256 GB RAM |
| *Software* | |
| Operating System | Ubuntu 22.04 LTS |
| Deep Learning Framework | PyTorch 2.4.0 |
| CUDA Toolkit | 12.4 |
| CL Baseline Library | Avalanche |
| ANN Search Library | HNSWlib |
| DINOv2 Model Source | PyTorch Hub (facebookresearch/dinov2) |

### C.3 HYPERPARAMETER INTERPLAY AND PRACTICAL TUNING GUIDANCE

The tables of hyperparameters in this appendix provide specific settings for each benchmark. However, a deeper understanding of the SORN framework comes from recognizing the interplay between its core parameters. These parameters are not independent knobs but are part of a dynamic system governing the model's balance between plasticity, stability, and efficiency. This section provides practical intuition and a recommended strategy for tuning SORN on a new problem.

The parameters can be conceptually grouped into three functional categories: (1) Structural Plasticity, (2) Homeostasis and Memory, and (3) Adaptation Speed.

**1. The Interplay of Structural Plasticity Parameters ($\alpha$ and $k$).** The network's topology is primarily governed by the novelty threshold scale ($\alpha$) and the number of k-WTA winners ($k$). These two parameters operate in a delicate balance to control the granularity of the learned representation.

- $\alpha$ **(Novelty Threshold Scale):** This is the most critical parameter controlling the network's growth rate. It directly defines a resonator's "personal space." A **lower** $\alpha$ makes resonators more "vigilant," leading to a higher rate of growth as more inputs are deemed novel. This results in a fine-grained tessellation of the feature space, creating many specific resonators. A **higher** $\alpha$ makes resonators more "tolerant," reducing the growth rate and encouraging the formation of more general, broadly-tuned resonators.

- $k$ **(Number of Winners):** This parameter defines the size of the activated ensemble and thus the richness of the Sparse Distributed Representation (SDR). Its interaction with $\alpha$ is subtle but important. A larger $k$ means that for any given input, a wider neighborhood of resonators is activated and refined. This tends to create a smoother, more distributed representation. Consequently, a slightly **lower** $\alpha$ might be required to differentiate nearby but distinct concepts when $k$ is large. Conversely, a small $k$ (e.g., $k = 3$) creates a more localized representation, which might pair well with a slightly **higher** $\alpha$ to prevent excessive growth from minor input variations.

The trade-off is between representational capacity and efficiency. A low-$\alpha$, high-$k$ setting will create a large, rich, and highly descriptive network, at the cost of memory and computation. A high-$\alpha$, low-$k$ setting results in a more compact and efficient "gist" model.

**2. The Interplay of Homeostasis and Memory Parameters ($\eta_u$ and $\theta_{prune}$).** If structural plasticity controls the birth of resonators, homeostasis controls their death, thereby defining the network's memory retention characteristics.

- $\eta_u$ **(Utility Learning Rate):** This parameter sets the timescale for utility tracking. A **low** $\eta_u$ (e.g., 0.001) creates a long-term memory filter; a resonator's utility reflects its activation history over thousands of samples. This makes the system robust to short-term fluctuations in the data distribution. A **higher** $\eta_u$ makes the utility more responsive to recent history, allowing the network to more quickly prune resonators that have recently become obsolete.
- $\theta_{prune}$ **(Utility Pruning Threshold):** This is the "survival threshold." Its value should be interpreted in the context of the total probability budget, $k$, as proven in Proposition 1. An approximate lower bound for the average activation probability of a surviving resonator is $\theta_{prune}$. Therefore, setting $\theta_{prune}$ is equivalent to defining the minimum "usefulness" a concept must have to be retained. A **very low** $\theta_{prune}$ allows the network to maintain "niche" resonators that represent rare but potentially important concepts. A **higher** $\theta_{prune}$ enforces a more parsimonious representation, keeping only the most salient and frequently activated concepts.

The synergy is clear: a low $\eta_u$ paired with a low $\theta_{prune}$ creates a network with very strong long-term memory, suitable for environments where old knowledge remains relevant. A higher $\eta_u$ and $\theta_{prune}$ create a more adaptive network that prioritizes recent information.

**3. Adaptation Speed (Learning Rates $\eta_k$ and $\eta_c$).** These parameters control the speed at which resonator positions and their associations are updated.

- $\eta_k$ **(Resonator Learning Rate):** Controls how quickly resonators move towards input data. A higher value leads to faster convergence but can cause instability if the data is noisy.
- $\eta_c$ **(Association Learning Rate):** Controls how quickly temporal associations are formed. A higher value allows for rapid learning of new sequences, while a lower value builds a more stable, long-term predictive model.

These are generally less sensitive than the structural parameters and can often be set to small, stable values (e.g., 0.01-0.05).

**A Practical Tuning Strategy.** Based on this understanding, we recommend the following hierarchical tuning process for applying SORN to a new dataset:

1. **Step 1: Set Sparsity Level.** Choose a value for $k$ based on the expected complexity of the data. $k = 3$ or $k = 5$ is a robust starting point.
2. **Step 2: Calibrate Growth Rate.** Fix all other parameters to reasonable defaults (e.g., $\eta_k = 0.02, \eta_u = 0.001, \theta_{prune} = 0.01$). Tune $\alpha$ to achieve a desired network size. Monitor the number of resonators ($N_t$) during a training run. If it grows uncontrollably, increase $\alpha$. If it barely grows, decrease $\alpha$.
3. **Step 3: Adjust Memory Retention.** Once the network size is stable, inspect the utility values. If you need to retain more niche concepts, lower $\theta_{prune}$. If the network seems cluttered with redundant resonators, slightly increase $\theta_{prune}$ or $\eta_u$.
4. **Step 4: Fine-tune Adaptation Speed.** Finally, adjust $\eta_k$ and $\eta_c$ if needed, based on the learning curves for accuracy and prediction tasks.

This structured approach transforms hyperparameter tuning from a brute-force search into an interpretable process of sculpting the model's lifelong learning behavior.

## C.4 HYPERPARAMETER SENSITIVITY ANALYSIS

To validate the robustness of the SORN framework, we conducted a sensitivity analysis on its two most critical hyperparameters governing structural plasticity: the novelty threshold scale, $\alpha$, and the number of k-WTA winners, $k$. We evaluated the model's performance on the Split CIFAR-100 benchmark while varying one parameter and keeping all others fixed to their optimal values as reported in Table 6.

The results of this analysis are presented in Figure 2. The plots clearly demonstrate that while extreme values can degrade performance, there exists a broad and stable range around the optimal settings where both average accuracy and forgetting remain near-optimal.

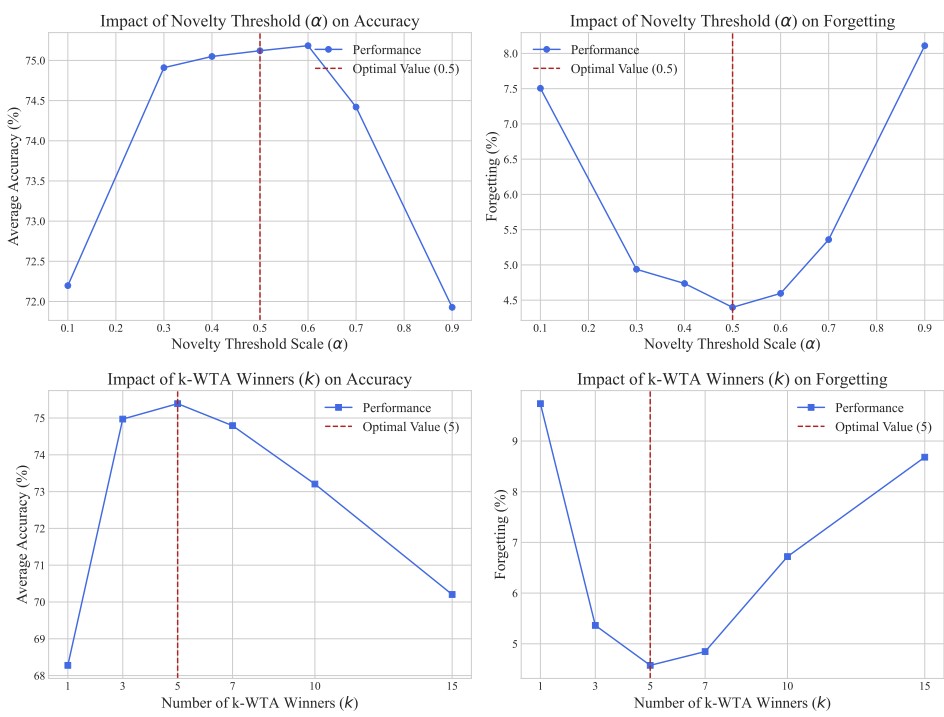

Figure 2: **Sensitivity analysis of key hyperparameters on Split CIFAR-100.** The top row shows the effect of varying the novelty threshold scale ($\alpha$), while the bottom row shows the effect of varying the number of k-WTA winners ($k$). Performance (Average Accuracy and Forgetting) is stable across a wide range of values around the optimal settings (indicated by the dashed red line), confirming the model's robustness.

Specifically, for the novelty scale $\alpha$, values between 0.4 and 0.6 yield top-tier performance, indicating that the novelty-gating mechanism is not dependent on a single, precisely tuned value. Similarly, for the number of winners $k$, values from 3 to 7 result in strong and stable outcomes. Performance degrades gracefully outside these ranges. For instance, a $k = 1$ setting (equivalent to 1-WTA) leads to a significant drop in accuracy and a sharp rise in forgetting, empirically validating our architectural choice. Overall, this analysis supports the claim made in the main paper that SORN is not overly sensitive to its hyperparameter settings, enhancing its practicality for real-world applications.

## D APPENDIX D: ADDITIONAL EXPERIMENTAL RESULTS & ANALYSIS

This section provides supplementary experimental results that further validate the robustness, stability, and efficiency of the SORN framework.

## D.1 TASK ORDER ROBUSTNESS ANALYSIS

A potential confounding factor in continual learning benchmarks is the sensitivity of a model to the specific sequence in which tasks are presented. A truly robust model should perform consistently regardless of the task order. To verify that SORN's superior performance is not an artifact of a favorable task sequence, we repeated the Split CIFAR-100 experiment using five different random permutations of the 20 tasks.

Figure 3 presents the average performance and standard deviation of SORN and key baselines across these five runs. The results clearly demonstrate SORN's stability. While replay-based methods like VERSE show slightly higher variance, SORN maintains both high accuracy and the lowest forgetting rate, with minimal deviation across runs. This indicates that its self-organizing principles lead to a learning process that is fundamentally less dependent on the order of incoming data, a critical attribute for real-world lifelong learning agents.

**Robustness to Task Order on Split CIFAR-100**

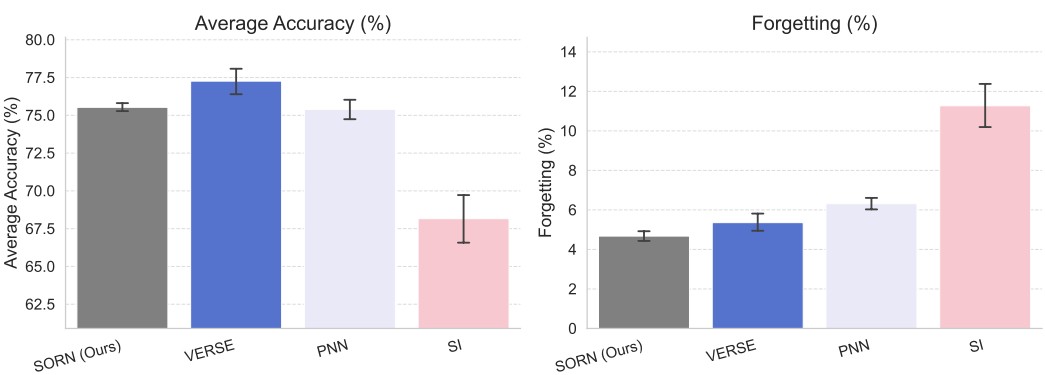

Figure 3: **Robustness to task order variation on Split CIFAR-100.** The bars show the mean performance (Average Accuracy and Forgetting) across five runs with different random task orders. The error bars represent the standard deviation. SORN achieves competitive accuracy and state-of-the-art forgetting with the smallest standard deviation in both metrics, highlighting its superior stability and robustness to the sequence of learning experiences.

## D.2 LEARNING PROCESS CURVES

While final performance metrics provide a summary of a model's capabilities, observing the evolution of performance throughout the learning process offers deeper insights into its forgetting dynamics. In Figure 4, we plot the average accuracy on all seen tasks as a function of the number of tasks learned.

The curve for an ideal continual learning agent would be a flat line, close to the offline upper bound. The analysis shows that SORN's performance curve is significantly flatter than all other baselines, visually confirming its exceptional ability to retain knowledge. While other methods exhibit a clear and steady decline in average accuracy as new tasks are introduced, SORN's accuracy degrades at a much slower rate. This provides strong visual evidence for the effectiveness of its homeostatic and self-organizing plasticity mechanisms in mitigating catastrophic forgetting over an extended learning period.

## D.3 MEMORY FOOTPRINT GROWTH CURVE

For a lifelong learning system to be practical, its resource consumption must be scalable. Figure 5 provides a direct comparison of the total memory footprint of SORN, a replay-based method, and the Progressive Neural Network (PNN) as they learn more tasks.

The results, plotted on a logarithmic scale to accommodate the vast differences, highlight the profound efficiency of SORN's design. Replay-based methods maintain a constant, moderate overhead. In stark contrast, PNN's approach of allocating a new network for each task results in unsustainable

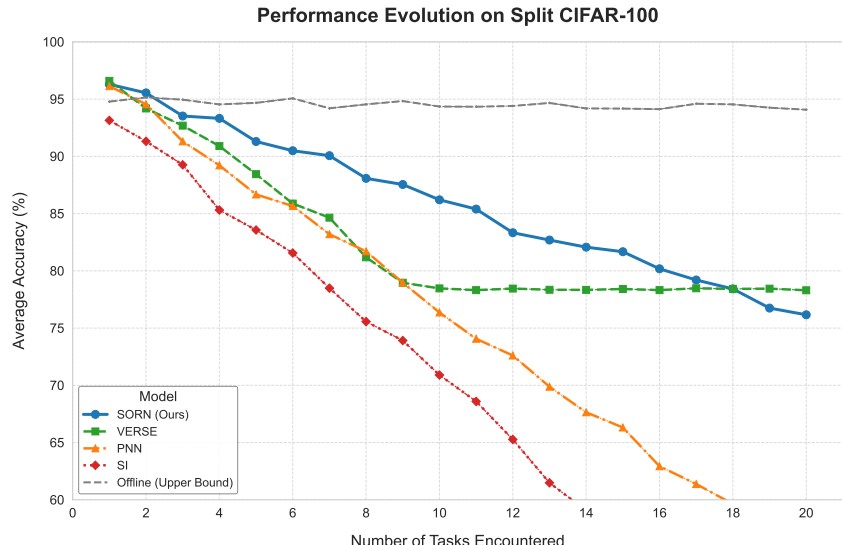

Figure 4: **Evolution of average accuracy during continual learning on Split CIFAR-100.** The plot tracks the average accuracy across all previously seen tasks as the model progresses through the 20-task sequence. The relative flatness of SORN's curve compared to the steeper declines of the baselines provides a clear visualization of its superior knowledge retention and resistance to catastrophic forgetting.

linear growth in its memory footprint, quickly becoming impractical. SORN charts a middle path of exceptional efficiency: its memory usage grows in a modest, sub-linear fashion as its resonator set expands to accommodate new concepts. This demonstrates that SORN not only excels in performance but also represents a highly scalable and resource-efficient architecture for true lifelong learning.

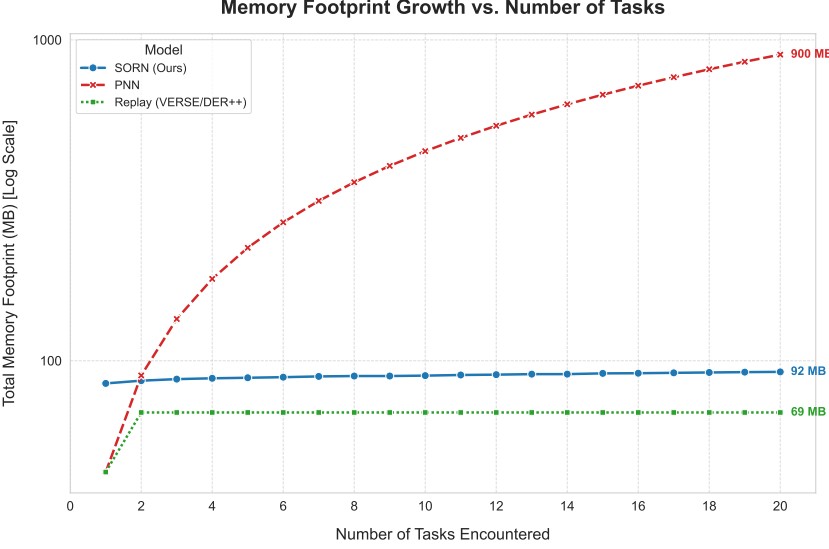

Figure 5: **Total memory footprint growth as a function of learned tasks.** The y-axis is on a logarithmic scale to clearly visualize the orders-of-magnitude difference in scalability. SORN's memory usage exhibits a controlled, sub-linear growth, confirming its efficiency. This stands in sharp contrast to the unsustainable linear growth of PNN, establishing SORN as a far more scalable solution for long-term continual learning.

# E LLM USAGE STATEMENT

We utilized a large language model (LLM) solely for the purpose of refining grammar, punctuation, and phrasing in this manuscript. The LLM was not used for generating any of the core scientific content, such as the methodology, experiments, or conclusions presented herein.

---

**Algorithm 1** The SORN Learning Algorithm for a Single Timestep

---

1: **Input:** Current network state $S_t = (R_t, C_t, U_t, \Theta_t)$, input embedding $e(t)$, previous activation vector $z(t-1)$.
2: **Hyperparameters:**
3:     $k$: number of winners in k-WTA.
4:     $\alpha$: novelty threshold scaling factor.
5:     $\eta_k$: base learning rate for resonators.
6:     $\beta$: sharpness parameter for distance-weighted learning.
7:     $\eta_c$: learning rate for association matrix.
8:     $\eta_u$: learning rate for utility tracking.
9:     $\theta_{prune}$: utility threshold for pruning.
10:     $\theta_{init}$: default novelty threshold for the first resonator.
11:     $P_{prune}$: pruning check period (in timesteps).

12: Let $N_t = |R_t|$ be the current number of resonators.
13: Initialize current activation vector $z(t) \in \{0, 1\}^{N_t}$ to all zeros.

14: **if** $N_t < k$ **then**                                    ▷ Handle state where not enough resonators exist for k-WTA.
15:     **goto** GROWTH.
16: **end if**

17:                                                    ▷ — **Activation Phase** —
18: Compute distances $d_i = \|e(t) - k_i\|_2$ for all $i \in \{1, \ldots, N_t\}$.
19: Let $\mathcal{W}(t)$ be the set of indices of the $k$ resonators with the smallest distances.
20: Set $z_i(t) = 1$ for all $i \in \mathcal{W}(t)$.
21: Let $win_1 = \arg\min_{i \in \mathcal{W}(t)} d_i$ be the index of the primary winner (closest resonator).

22:                                                    ▷ — **Novelty Detection Phase** —
23: **if** $d_{win_1} > \theta_{win_1}$ **then**                    ▷ Input is novel if it falls outside the primary winner's radius.
24:     **goto** GROWTH.
25: **else**
26:     **goto** REFINEMENT.
27: **end if**

28:  **GROWTH Procedure:**
29: Create a new resonator $k_{N_t+1} \leftarrow e(t)$.
30: Add $k_{N_t+1}$ to the resonator set: $R_{t+1} \leftarrow R_t \cup \{k_{N_t+1}\}$.
31: **if** $N_t > 0$ **then**                          ▷ Growth triggered by novelty or insufficient resonators.
32:     Initialize utility: $U_{N_t+1} \leftarrow \text{mean}(U_t)$.
33:     Initialize novelty threshold: $\theta_{N_t+1} \leftarrow \alpha \cdot \|e(t) - k_{win_1}\|_2$.            ▷ $k_{win_1}$ is defined.
34: **else**                                          ▷ This is the very first resonator in the network.
35:     Initialize utility: $U_{N_t+1} \leftarrow 1.0$.
36:     Initialize novelty threshold: $\theta_{N_t+1} \leftarrow \theta_{init}$.
37: **end if**
38: Augment association matrix $C_t$ and utility vector $U_t$ with a new row/column of zeros.
39: $N_{t+1} \leftarrow N_t + 1$.
40: **goto** ASSOCIATIVE_LEARNING.

41:  **REFINEMENT Procedure:**
42: Compute softmax-based weights for winners: $w_i = \frac{\exp(-\beta d_i^2)}{\sum_{j \in \mathcal{W}(t)} \exp(-\beta d_j^2)}$ for each $i \in \mathcal{W}(t)$.
43: Update winning resonators: $k_i(t+1) \leftarrow (1 - \eta_k w_i)k_i(t) + \eta_k w_i e(t)$ for each $i \in \mathcal{W}(t)$.
44: Non-winning resonators remain unchanged: $k_j(t+1) \leftarrow k_j(t)$ for $j \notin \mathcal{W}(t)$.
45: $R_{t+1} \leftarrow R_t$.

46:  **ASSOCIATIVE_LEARNING Procedure:**
47: Update association matrix with stabilized Hebbian rule:
48: $C_{t+1} \leftarrow C_t + \eta_c(z(t)z(t-1)^T - \text{diag}(z(t))C_t)$.

49:                                                    ▷ — **Homeostasis and Pruning Phase** —
50: Update all resonator utilities: $U_{t+1} \leftarrow (1 - \eta_u)U_t + \eta_u z(t)$.
51: Update all novelty thresholds $\Theta_t$ based on new resonator positions. For each $i$, $\theta_i \leftarrow \alpha \cdot \min_{j \neq i} \|k_i - k_j\|_2$.
52: **if** $t \pmod{P_{prune}} == 0$ **then**
53:     Identify resonators to prune: $\mathcal{P} = \{i \mid U_i(t+1) < \theta_{prune}\}$.
54:     **if** $|\mathcal{P}| > 0$ and $(N_t - |\mathcal{P}|) \geq k$ **then**              ▷ Ensure at least k resonators remain.
55:         Remove resonators with indices in $\mathcal{P}$ from $R, C, U, \Theta$.
56:     **end if**
57: **end if**

58: **Output:** Updated state $S_{t+1}$, current activation vector $z(t)$.

---

1728
1729
1730
1731
1732
1733
1734
1735
1736
1737
1738
1739
1740
1741
1742
1743

---

**Algorithm 2** SORN Supervised Inference Protocol

---

**Part A: Training (Online Label Association)**
1: **procedure** ASSOCIATELABEL($S_t, e(t), y(t)$)
2:     Let $S_t = (R_t, C_t, U_t, \Theta_t, L_t)$ be the full network state including label vectors.
3:     Determine winning resonator set $\mathcal{W}(t)$ for input $e(t)$.
4:     Let $\mathbf{y}_{oh}(t)$ be the one-hot vector for label $y(t)$.
5:     **for** each resonator index $i \in \{1, \ldots, N_t\}$ **do**
6:         **if** $i \in \mathcal{W}(t)$ **then**
7:             $L_i(t+1) \leftarrow (1 - \eta_L)L_i(t) + \eta_L \cdot \mathbf{y}_{oh}(t)$.
8:         **else**
9:             $L_i(t+1) \leftarrow L_i(t)$.
10:         **end if**
11:     **end for**
12:     **return** Updated label vectors $L_{t+1}$.
13: **end procedure**

**Part B: Testing (Inference)**
14: **procedure** PREDICTCLASS($S_{final}, e_{test}$)
15:     Let $S_{final} = (R, C, U, \Theta, L)$ be the final trained network state.
16:     Determine winning resonator set $\mathcal{W}(test)$ for input $e_{test}$.
17:     Initialize score vector $S_{pred} \in \mathbb{R}^C$ to all zeros.
18:     **for** each resonator index $i \in \mathcal{W}(test)$ **do**
19:         $S_{pred} \leftarrow S_{pred} + L_i$.
20:     **end for**
21:     $\hat{y} \leftarrow \arg\max_j (S_{pred})_j$.
22:     **return** Predicted class $\hat{y}$.
23: **end procedure**

---
