# OpenReview forum: "Self-Organizing Resonant Network"
_ICLR.cc/2026/Conference — Submitted to ICLR 2026_

### Official Review · Reviewer_LrGA · 2025-10-24

**Soundness:** 2
**Presentation:** 2
**Contribution:** 3
**Rating:** 4
**Confidence:** 2

**Summary:**

This paper proposes a biologically inspired continual learning framework (SORN). Instead of backpropagation use two complementary mechanisms:
(1) Novelty-Gated Structural Plasticity, which creates or prunes “resonator” neurons as new concepts appear,
(2) Stable Hebbian Synaptic Plasticity, which learns sparse inter-concept associations under homeostatic constraints.

SORN operates in a fixed feature space (e.g., DINOv2 embeddings). The paper provides theoretical analysis. The evaluation is performed on several continual learning benchmarks.

**Strengths:**

1. The paper revisits biologically motivated, non-gradient-based learning — an underexplored but important area.
2. The authors provide an analysis of stability and convergence.

**Weaknesses:**

1. SORN’s stability crucially depends on pre-trained DINOv2 embeddings. This outsources representation learning to an external model, weakening the claim that it solves continual learning “end-to-end.” The system’s success may largely stem from DINOv2’s high-quality representations rather than from SORN’s mechanisms.
2. The backprop-based baselines are trained end-to-end from scratch, whereas SORN uses a fixed high-capacity pretrained model. This unfairly advantages SORN in stability and efficiency metrics. An alternative comparison would freeze the encoder for baselines as well.
3. SORN is compared to outdated baselines. Since the authors use pretrained model, they should compare to methods of similar spirit like FeTrIL [1], FeCAM [2], or H-Prompts [3].
4. The paper is quite hard to read clearly.

[29] Petit, G., Popescu, A., Schindler, H., Picard, D., Delezoide, B.: Fetril: Feature translation for exemplar-free class-incremental learning. In: Proceedings of the IEEE/CVF winter conference on applications of computer vision. pp. 3911–3920 (2023)
[10] Goswami, D., Liu, Y., Twardowski, B., van de Weijer, J.: Fecam: Exploiting the heterogeneity of class distributions in exemplar-free continual learning. In: Advances in Neural Information Processing Systems (NeurIPS) (2023)
[42] Zhanxin Gao, Jun CEN, X.C.: Consistent prompting for rehearsal-free continual learning (2024)

**Questions:**

Can the model operate effectively when the encoder is fine-tuned jointly (e.g., partially trainable features)?

---

### Official Review · Reviewer_DasP · 2025-10-26

**Soundness:** 2
**Presentation:** 3
**Contribution:** 3
**Rating:** 4
**Confidence:** 4

**Summary:**

This paper proposes the Self-Organizing Resonant Network (SORN) for continual learning.
The approach first uses a fixed, pre-trained encoder to extract features, and then applies a non-backpropagation learning mechanism that self-organizes. The network dynamically adds or prunes neurons based on a utility-driven novelty threshold and updates connections using a stabilized hebbian rule.
Experiments on standard continual learning benchmarks compare SORN against regularization-based, replay-based, and dynamic architecture methods, and show less forgetting, and better performance. The paper also performs ablations to examine the effect of the components of their proposed method, the fixed encoder, and the update rule.

**Strengths:**

1. The paper is well-motivated and well-written. The method is explained thoroughly, and the notations are easy to follow.

2. The paper shows strong experimental rigor, including careful hyperparameter tuning for both the proposed method and the baselines, an ablation study of key components, and sensitivity analyses for the new hyperparameters introduced by the approach.

3. The paper covers the related literature well, and includes a wide range of baselines, both classical and recent,i n the experiments to compare with their method

**Weaknesses:**

1. The use of the encoder is a fundamental part of the proposed method in the paper. However, a key challenge in continual learning is actually for the model to learn the representations over time. The paper needs to disentangle the effect of the use of the encoder and the power of the design and structure of the network. For instance, how does a backpropagation-based network work in comparison to SORN, if its input is the encodings derived by the encoder? Furthermore, more investigation into the role of the encoder would be helpful. For instance, how does the performance change if we use other encoders/foundation models to extract features?


2. Performance metrics should be reported with the number of seeds run for each method, and confidence intervals  and preferably statistical tests to show significant differenc,e especially between the methods that have close performance values (e.g., close forgetting values) with SORN. Since confidence intervals are not specified, this is unclear.


3. The theoretical prepositions assume stationary input. While it is very valuable to examine the properties of SORN, like network size, boundedness, and convergence in a stationary setting, the fundamental difference in CL is that the input can be non-stationary. How do you expect these properties to change under non-stationarity?


4. The paper rightly emphasizes task-agnostic continual learning as one of SORN’s main advantages. However, the reported metrics still depend on predefined task boundaries. Although identifying more suitable metrics for task-agnostic evaluation remains an open challenge, including additional measures such as online learning curves would make the evaluation more aligned with the paper’s stated goals.

Minor comments:

Add a reference for some of the keywords, like loss of plasticity

Add the reference for CHNN in line 332

Add a reference to the appendix in the paragraph around line 297, referring to the hyperparameter sensitivity analysis section

Add the reference in the text to tables 1 and 2 when explaining the results

**Questions:**

1. How do you decouple the power of the encoder from your network structure? How does SORN compare to methods that use the same feature extraction method, but, e.g., backpropagation-based networks?

2.  Why did you choose the DINOV2 encoder? How do the experiment results change in the case of the use of other feature extractors?

3. What is the number of seeds run, and what are the confidence intervals?

---

### Official Review · Reviewer_ZQ9h · 2025-11-01

**Soundness:** 2
**Presentation:** 4
**Contribution:** 3
**Rating:** 6
**Confidence:** 4

**Summary:**

The paper introduces the Self-Organizing Resonant Network, a novel paradigm for continual learning that operates without backpropagation except for the pre-trained feature extractor. SORN addresses catastrophic forgetting through two biologically-inspired principles: Novelty-Gated Structural Plasticity to dynamically create new "resonators" for novel concepts, and Stable Hebbian Synaptic Plasticity to learn sparse associations. This approach enables the network to achieve state-of-the-art resistance to catastrophic forgetting in various continual learning benchmarks, especially in non-stationary, task-agnostic environments.

**Strengths:**

The paper is very clearly presented and the proposed method is easy to understand. The approach is significant because of its non-backpropagation, bio-inspired update rule rooted in Hebbian learning. It thus offers potential contributions to both the neuroscience and continual learning communities. This novel architecture delivers strong empirical performance, demonstrating state-of-the-art resistance to catastrophic forgetting against leading baselines. Crucially, the work is supported by a robust theoretical analysis that formally clarifies the network’s convergence properties and capacity scaling, moving the contribution beyond a purely empirical result.

**Weaknesses:**

1. My main concern about this work is whether the comparison to other CL methods in Table 1 is fair, given that the current method uses an advanced pre-trained visual encoder (DINOv2). See questions below.

2. If possible, I would suggest adding a brief discussion of the limitations of the current method and potential directions for future work.

3. The proposed method involves a range of different hyperparameters, and the settings vary considerably across datasets. This raises concerns about whether the method could scale to real-world scenarios with continuous distributional shifts.

**Questions:**

1. Could the authors clarify which baselines also use a fixed pre-trained encoder and whether they use the same vision encoder? Is it possible to compare to these baselines after replacing their visual encoders with a frozen pre-trained DINOv2?

2. The hyperparameter settings appear to differ substantially across datasets (Tables 5–8). How are these hyperparameters tuned? The authors discuss the sensitivity of performance to two key parameters, k and α, but it would be helpful to extend this analysis to different datasets and to other parameters that are assigned different values across datasets.

---

### Official Review · Reviewer_8Gek · 2025-11-01

**Soundness:** 1
**Presentation:** 3
**Contribution:** 2
**Rating:** 2
**Confidence:** 4

**Summary:**

This paper proposes a self-organizing, dynamically growing and shrinking networks architecture for continual learning that is validated on several common CL benchmarks.

**Strengths:**

The paper gives a very detailed and well-reasoned account of the SORN model, taking great care to explain the reasonings behind design choices. The model itself is interesting and novel, although the merit of the individual concepts is hard to verify. Comparisons on common benchmarks are given w.r.t. to a wide number of baseline algorithms using a public reference library.

**Weaknesses:**

I have grave concerns concerning a number of points that often come up in CL research. I have formulated questions w.r.t. each of these, and I may adapt my score depending on the answers:
- fairness of comparison: if you compare SORN to other algorithms, it is unfair to use feature encoding for SORN only
- hyper-parameter selection: SORN critically depends on a rather large set of hyper-parameters. It looks like they are selected by grid-search  based on a whole CL experiment with all tasks. That is strongly illegal in CL, since you would use knowledge of *all* tasks of the benchmark to select hyper-parameters. The point of CL is *not* having access to future data. So if you select hyper-parameters, you can do it on task 1 data but not more.
- hyper-parameter selection for baselines: it is highly questionable to carefully tune hyper-parameters for the own algorithm while using default parameters for the baselines. We know that, e.g., \lambda in EWC play are crucial role, and so do buffer sizes etc. in GEM or iCarL.

**Questions:**

- when comparing to other CL baselines: do the other algorithms work on the pre-encoded feature space or on the raw data? Because if the latter is true, how is this comparison fair?
- how are the hyper-parameters for SORN fixed? In the appendix you mention grid search for (k, \alpha), what about the others? Please explain this process in more detail? What validation data are you using for this grid-search, is it the whole dataset that is tested after training on all tasks, or just a subset, e.g., task 1?
-  is the fixed feature extractor (dinoV2) adapted to a particular dataset? Or do you just re-scale all images to the encoders' fixed input size and always use the same one?
- please comment on the following statement: the fixed feature encoder makes each problem essentially linearly separable, and therefore the inherent difficulty of the different benchmarks is irrelevant
- can you show the performance of a simple linear classifier on task 1 from each benchmark as preprocessed by the feature encoder? Just to demonstrate the inherent difficulty of the benchmarks?
- how are parameters to baseline algorithms chosen? Do you use default parameters or do you optimize those as well?

---

### Meta-Review · Area_Chair_rLVk · 2026-01-07

**Summary:**

This paper proposes the SORN model for continual learning (CL) and validates it on several common CL benchmarks. The reviewers raised common concerns regarding the fairness of the comparisons with baselines, due to the use of an advanced pretrained visual encoder in the proposed SORN. Additional major concerns include improper hyperparameter selection, the lack of performance metrics and confidence intervals in the reported results, and the use of outdated baselines. The authors did not provide a rebuttal. Based on the reviews and the reviewers’ scores, this paper is recommended for rejection.

**Reviewer Concerns:**

There is no rebuttal.

**Reviewer Scores:**

N/A, no rebuttal.

---

### Decision · Program_Chairs · 2026-01-26

Reject